# Microchannelled alkylated chitosan sponge to treat noncompressible hemorrhages and facilitate wound healing

Xinchen Du[1], Le Wu[1], Hongyu Yan[1], Zhuyan Jiang[2], Shilin Li[1], Wen Li[1], Yanli Bai[1], Hongjun Wang[3], Zhaojun Cheng[4], Deling Kong[1✉], Lianyong Wang[1✉] & Meifeng Zhu [1✉]

Developing an anti-infective shape-memory hemostatic sponge able to guide in situ tissue regeneration for noncompressible hemorrhages in civilian and battlefield settings remains a challenge. Here we engineer hemostatic chitosan sponges with highly interconnective microchannels by combining 3D printed microfiber leaching, freeze-drying, and superficial active modification. We demonstrate that the microchannelled alkylated chitosan sponge (MACS) exhibits the capacity for water and blood absorption, as well as rapid shape recovery. We show that compared to clinically used gauze, gelatin sponge, CELOX™, and CELOX™-gauze, the MACS provides higher pro-coagulant and hemostatic capacities in lethally normal and heparinized rat and pig liver perforation wound models. We demonstrate its anti-infective activity against *S. aureus* and *E. coli* and its promotion of liver parenchymal cell infiltration, vascularization, and tissue integration in a rat liver defect model. Overall, the MACS demonstrates promising clinical translational potential in treating lethal non-compressible hemorrhage and facilitating wound healing.

[1] College of Life Sciences, Key Laboratory of Bioactive Materials (Ministry of Education),Tianjin Center Hospital of Obstetrics and Gynecology, State Key Laboratory of Medicine Chemical Biology, Nankai University, Tianjin, China. [2] Department of Orthopedics, The Second Hospital of Tianjin Medical University, Tianjin, China. [3] Department of Biomedical Engineering, Stevens Institute of Technology, Hoboken, NJ, USA. [4] Shenzhen Traditional Chinese Medicine Hospital, Shenzhen, China. ✉email: kongdeling@nankai.edu.cn; wly@nankai.edu.cn; zhumeifeng2013@163.com

Hypotension and multi-organ failure caused by massive blood loss often results in high mortality in civilian and military populations[1,2]. So, rapid and efficient hemorrhage control is of paramount importance in such scenarios. The body's natural coagulation cascade process is activated in response to bleeding, but, incapable of timely stopping severe hemorrhage from a deep and noncompressible perforation wound in the absence of shape-memory hemostats[3,4]. Thus, the development of shape-memory hemostats is urgently needed. In general, ideal shape-memory hemostats should possess several properties, including a highly interconnected porous structure, active coagulation, strong anti-infection activity, biocompatibility, biodegradability, ready availability, low weight, and low cost[4–7]. Notably, an interconnected porous structure permits fluid to flow freely in and out of hemostats, which allows hemostats to be fixed by draining off the free water and promotes fast recovery to their initial shapes by absorbing the fluid[6]. Rapid shape recovery timely exerts pressure on the wound, leading to effective hemorrhage control[6,8]. Moreover, hemostats left in the injury site and used in directly guiding in situ tissue regeneration are more practical for clinical application[9].

Until now, several shape-memory hemostats have been developed, and some have been applied in clinical practice[10–13]. For instance, the XStat™ device composed of multiple compressed cellulose sponges was shown to rapidly expand to fill and exert pressure on the wound to control hemorrhage[10]. However, it took much more time to take out each sponge from the wound bed due to its nondegradable property, which may cause patient discomfort[8]. Moreover, such a sponge lacking a highly interconnected porous structure was incapable of guiding tissue repair. Many shape-memory polymer foams as hemostats have been applied to treat noncompressible hemorrhage and exhibited a certain degree of hemostatic ability[11–13]. However, they displayed limited absorption of blood and required decades of seconds to restore their shapes, which may cause the prolongation of hemostatic time and more blood loss[5]. Injectable cryogels with high blood absorbability and rapid-shape recovery capacity have also been developed for the treatment of noncompressible hemorrhage[6,14,15]. The hemostatic effect of these materials was achieved by restoring shape and applying mechanical compression to the wound. Shape-recovery property mainly originates from the reversible change of porous structure[10–13]. However, the pores inside these hemostats generated by gas foaming or ice crystal removing methods possess low interconnectivity, which might slow down the blood flow into hemostats, resulting in weakened hemostatic efficiency. The effect of pore structure, especially interconnectivity, on hemostatic performance was usually ignored in the design and construction of hemostats[5,8,10,14]. Besides, some of these hemostats lacked strong active pro-coagulant and anti-infective properties, which may result in their failure to complete the hemostasis in a timely and effective way and in their inability to protect wounds from bacterial infection. Therefore, simultaneously regulating pore structure and active modification is expected to improve the hemostatic and anti-infective effects of these hemostats.

Incorporating a microchannel into three-dimensional (3D) constructs is a simple and controllable architectural feature, and is capable of promoting transport of nutrients, oxygen, and metabolites, host cell infiltration, vascularization, and integration with the surrounding tissue[16–19]. To create an embedded and hollow microchannel, the sacrificial fibrous template with a well-defined 3D architecture was first enclosed within a matrix material solution and later removed via external stimuli[20]. Such an approach showed better controllability and interconnectivity in pore structure than conventional pore-forming methods, including gas foaming and ice crystal removing[18]. Still,

developing shape-memory hemostats with a microchannel structure has not been previously investigated.

Chitosan (CS) has been used to prepare hemostats due to its inherent properties, such as biocompatibility, biodegradability, non-toxicity, anti-infection ability, hemostasis, and so forth[21,22]. Nevertheless, as mentioned above, its hemostatic and anti-infective properties were limited, especially in cases complicated by severe hemorrhage and bacterial infections[23]. Previous studies by our group and other groups have demonstrated that grafting hydrophobic alkyl chains onto a CS backbone could improve its hemostatic and anti-infective abilities, attributed to the strong hydrophobic interactions between the alkyl chains and the membranes of red blood cells (RBCs), platelets, and bacteria[23–26].

In this work, we incorporate a microchannel structure into a CS sponge and further modify it with hydrophobic alkyl chains. The MACSs achieve rapid shape recovery by absorption of water and blood. Compared with clinically used gauze, GS, CELOX™, and CELOX™-G, the MACSs demonstrate stronger pro-coagulant ability in vitro and hemostatic capacity in lethally normal and heparinized rat and normal pig liver perforation wound models. Moreover, the MACSs enable liver parenchymal cell infiltration, vascularization, and tissue integration in a rat liver defect model. All results indicate that the MACSs have the clinical translational capacity to provide effective treatment for potentially lethal noncompressible hemorrhages and wound healing.

## Results and discussion

**Fabrication and characterization of the MACSs**. According to our design criteria, the MACSs were fabricated by the procedure illustrated in Fig. 1a. First, the sacrificial PLA microfiber templates were printed by a 3D printer (Fig. 1b and Supplementary Fig. 1). Then, the templates were lyophilized after filling with a 4% (w/v) CS solution. The CS sponge with microchannel structure was obtained following complete removal of the PLA templates, which was confirmed by FTIR measurement (Supplementary Fig. 2). The resultant CS sponge was further grafted with hydrophobic alkyl chains to improve its pro-coagulant and anti-infective properties. The grafting was carried out via a highly efficient Schiff-base reaction between the amine group of CS and the aldehyde group of DA (Fig. 2a). The unstable C=N was converted into stable C–N using a reductant (NaCNBH₃). Compared to the N1s spectrum of the CS sponge, the appearance of C–N*H–C and reduction of the peak area of C–N*H₂ in the N1s spectrum of the alkylated CS sponge indicated the successful hydrophobic modification (Fig. 2b–d). The grafting degree of DA was $27.86 \pm 18.99\%$ (Fig. 2d).

Interconnected pores of the hemostatic sponge could endow itself with the ability to concentrate coagulation factors and rapidly recover to initial shape[5,10,27]. Moreover, they were able to provide a comfortable niche to support host cell infiltration, vascularization, and tissue ingrowth[28]. Micro-CT images showed that the alkylated CS sponges with different porosity (MACS-1/2/3) fabricated by a combination of the template leaching method and freeze-drying possessed a uniform microchannel structure with an increased microchannel density (Fig. 1c). The alkylated CS sponge (ACS) prepared by direct freeze-drying presented a dense structure. Furthermore, SEM images displayed a hierarchical porous structure including microchannel ($136.5 \pm 17.8\,\mu m$) and micropores ($8.3 \pm 0.8\,\mu m$) in the MACS-1/2/3 (Fig. 1d–f), while only micropores ($8.1 \pm 1.0\,\mu m$) randomly distributed throughout the ACS. The microchannel structure was highly interconnected and tunable and distributed uniformly across the MACS-1/2/3 (Supplementary Movies 1–3). However, the micropores distributed in the ACS showed a dense structure and low interconnectivity (Supplementary Movie 4). The interconnectivity

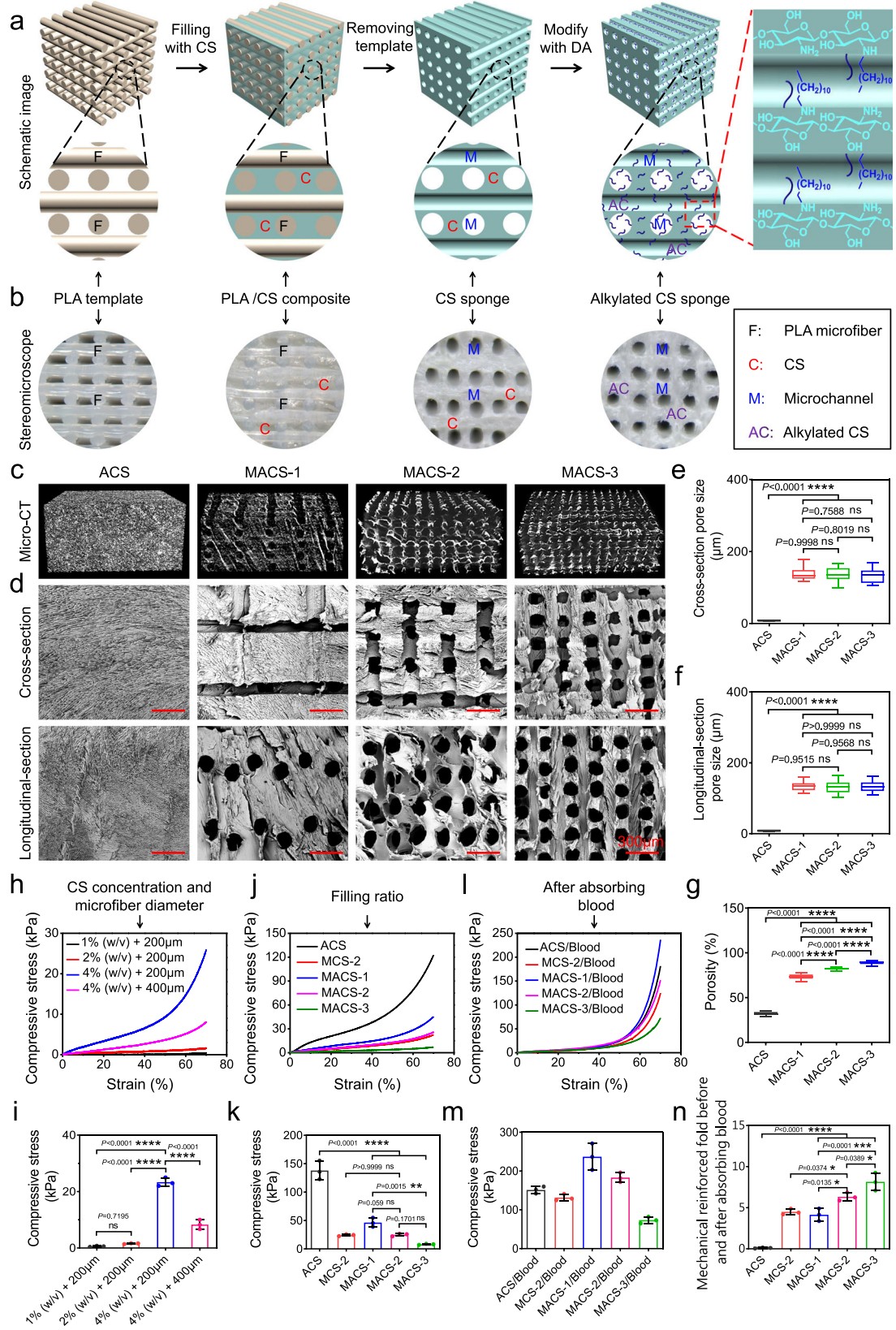

**Fig. 1 Fabrication and characterization of the MACSs with different porosity. a** Schematic illustration of the fabrication process of the MACSs.
**b** Stereomicroscopic images of the PLA microfiber template, CS/PLA composite, micro channeled CS sponge, and micro channeled alkylated CS sponge.
**c**, **d** Micro-CT and SEM images showing the macro and microstructure of the ACS and MACS-1/2/3. **e** The pore size of the ACS and MACS-1/2/3 in cross-section and longitudinal-section (pore size, $n = 16$). **f** The pore size of the ACS and MACS-1/2/3 in longitudinal-section (pore size, $n = 16$). **g** The porosity of the ACS and MACS-1/2/3 (porosity, $n = 25$). **h**, **i** Compressive stress-strain curves and compressive stress of the MACSs with different CS concentrations (1, 2, and 4% (w/v)) and PLA microfiber diameter (200 and 400 μm) ($n = 3$ independent samples). **j**–**m** Compressive stress-strain curves and compressive stress of the ACS, MCS-2, and MACS-1/2/3 before and after absorbing blood. **n** Mechanically reinforced folds of the ACS, MCS-2, and MACS-1/2/3 before and after absorbing blood ($n = 3$ independent samples). Data are expressed as mean ± SD. The significant difference was detected by one-way ANOVA with Tukey's multiple comparisons test. The 'ns' indicated no significant difference, *$P < 0.05$, **$P < 0.01$, ***$P < 0.001$, ****$P < 0.0001$.

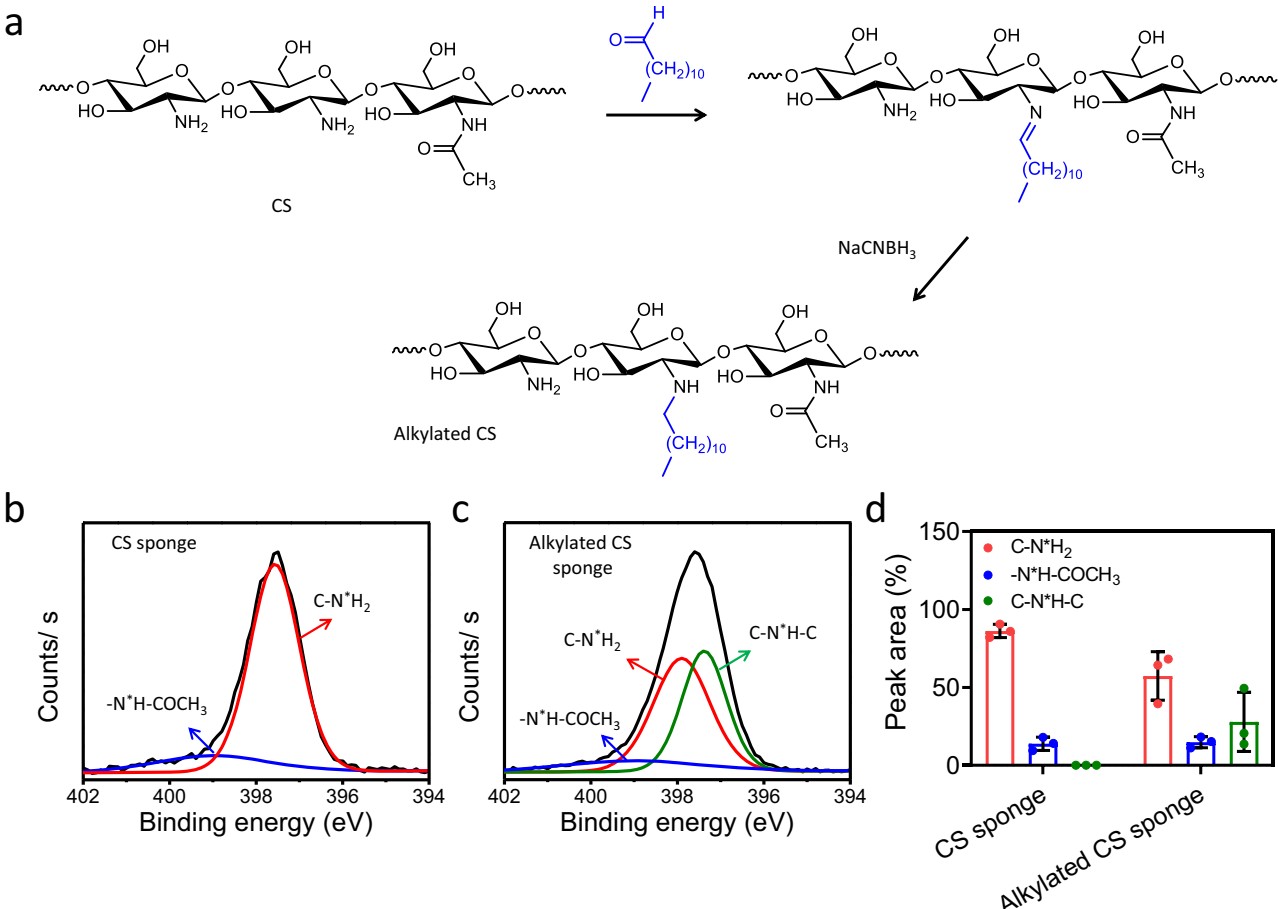

**Fig. 2 Chemical characterization of the MACSs. a** Modification of the CS sponge with DA in the presence of $NaCNBH_3$ as a reducing agent.
**b**, **c** Representative XPS spectra showing N1s peak of the CS and alkylated CS sponges. **d** The area of N1s peaks with different chemical states in the CS and alkylated CS sponges.

of the porous structure played a key role in accelerating hemostasis and guiding tissue regeneration, which usually was ignored in most previous studies[5,6,10,13]. The MACSs were expected to exhibit an obvious advantage in the treatment of noncompressible hemorrhage and in situ tissue regeneration in comparison with reported porous hemostats[5,6,10,14]. Accordingly, the porosity of the MACSs gradually increased from 73.2 ± 2.9 to 88.8 ± 1.6% with an increase in filling ratio of PLA microfiber, which were significantly higher than the 32.1 ± 1.9% of the ACS (Fig. 1g). Hemostats filled into the wound cavity should possess desirable mechanical strength to prevent their shape deformation caused by external stress from surrounding tissues, thereby providing durable compression on the bleeding site. We first examined the effect of CS concentration on the compressive stress of the MACSs. As the CS concentration increased from 1 to 4%

(w/v), the compressive stress was enhanced from 0.6 ± 0.2 to 23.0 ± 1.5 kPa (Fig. 1h, i). When the CS concentration was lower than 4%, the sponges could not maintain their shapes (Supplementary Fig. 3a). The CS solution with a concentration higher than 4% possessed higher viscosity (Supplementary Fig. 3b, c), and was difficult to be sucked into the gap of the PLA microfiber template under negative pressure. So, the 4% (w/v) CS solution was selected to fabricate the MACSs. Then, we explored the effect of the PLA microfiber diameter on compressive stress. With the increase of PLA microfiber diameter from 200 to 400 μm, the compressive stress decreased from 23.0 ± 1.5 to 8.0 ± 1.7 kPa (Fig. 1h, i). Next, we investigated the effect of the filling ratio of the PLA microfiber template on compressive stress. The compressive stress decreased from 46.2 ± 8.0 to 8.1 ± 0.9 kPa by increasing the filling ratio of the PLA microfiber template from 20

to 60% (Fig. 1j, k). Indeed, the compressive stress of the MACSs was significantly lower than the 138.0 ± 16.3 kPa of the ACS due to the incorporation of the microchannel structure. To better approach practical application, we further detected the compression stress of the sponges after absorbing blood. All the sponges exhibited reinforced mechanical strength (Fig. 1l, m), attributing to the formation of blood clots within the sponges. Both the CS and hydrophobic alkyl chains have been proven to facilitate blood clotting by promoting the adhesion and activation of platelets and the aggregation of RBCs[1,6,9,24]. The MACSs had a higher mechanically reinforced fold than the ACS (Fig. 1n). Also, the mechanically reinforced fold of the MACSs gradually enhanced with the increase in porosity (Fig. 1n). The MACSs with high porosity and large surface area could absorb more blood and facilitate the blood to fully contact with the matrix to form more blood clots. Also, the alkylated CS sponge (MACS-2) displayed an improved mechanically reinforced fold compared to the unmodified CS sponge (MCS-2) due to the introduction of hydrophobic alkyl chains (Fig. 1n).

**Water/blood absorbability of the MACSs.** The main hemostatic mechanism of expandable hemostats was mechanical compression on the bleeding site, which mainly resulted from water/blood-triggered shape recovery and volume expansion[1,5,14,29,27]. Thus, strong water/blood absorbability was indispensable for expandable hemostats. After absorbing the fluid (representative blood), the MACSs rapidly sank to the bottom of the container, while the ACS suspended in the fluid (Fig. 3a), revealing that the MACSs could absorb a higher volume of blood compared with the ACS. The maximum water and blood absorption capacity of the MACSs was significantly higher than that of the ACS and gradually improved with an increase in the porosity (Fig. 3b–e). Notably, the MACSs absorbed more water and blood than that of the ACS at the same time point (Fig. 3b, c). The water and blood absorption rate of the MACSs was higher than that of the ACS (Fig. 3f, g), which resulted from the increased number of microchannels. The more microchannels present, the higher the water and blood absorption rate. The blood absorbability of the MACSs was comparable with their water absorbability, whereas the blood absorbability of the ACS was obviously weaker than its water absorbability (Fig. 3b, c). The MACSs possessed a highly interconnected microchannel structure, which allowed water/blood to quickly penetrate into the inside of the MACSs (Fig. 3h, i). By contrast, the ACS with dense microporous structure inhibited the complete penetration of high-viscosity blood (Fig. 3i). We further stimulated the fluid absorption behavior of the sponges, as shown in Fig. 3j. We found that the fluid speed in the microchannels of the alkylated sponges (MACS-1/2/3) was higher than that in micropores of the ACS. The higher number of microchannels resulted in a larger area of distribution of the high fluid speed. The total fluid speed of the MACSs was higher than that of the ACS and gradually improved as the number of microchannels increased (Fig. 3k).

**Shape-memory property of the MACSs.** We further evaluated the water-triggered and blood-triggered shape-memory property of the MACSs and ACS. All sponges could be compressed and shape-fixed after squeezing out the free water (Fig. 4a, b). Upon absorbing the water, they could recover to their original shapes (Fig. 4a, c), giving a 100% recovery ratio. The recovery time (3.3 ± 0.6, 2.1 ± 0.1, and 1.7 ± 0.6 s) of the MACSs was significantly shorter than the 41 ± 3.6 s of the ACS (Fig. 4d and Supplementary Movies 5, 6). After absorbing blood, the shape-fixed MACSs could achieve full shape recovery (4.0 ± 1.0, 2.5 ± 0.5, and 2.1 ± 0.1 s) (Supplementary Movie 7). However, the ACS kept a

compressed shape and could not recover any further (Fig. 4b, d, f and Supplementary Movie 8). In addition, the shape recovery time of the MACSs after absorbing water/blood was significantly shorter than that of reported shape-memory hemostats (Fig. 4g). Indeed, a large number of studies have demonstrated that, compared to water, blood is more likely to prolong the shape recovery time of hemostats due to its higher viscosity[14,15]. In contrast, there was no significant difference in shape recovery time for the MACSs after the absorption of water and blood. This was attributed to the highly interconnected microchannel structure, which allowed the blood to freely penetrate into the interior of the sponges. The microporous structure inside the ACS and reported expandable hemostats generated by the removal of ice crystals and by gas foaming methods exhibited low interconnectivity, which inhibited the penetration of high-viscosity blood[5,10,11,14,15].

The microstructure of the compressed sponges after absorbing water and blood was further observed by SEM (Fig. 4h). In their original state, homogeneous and circular microchannels with gradient numbers distributed throughout the MACSs. The circle microchannels changed to flat channels under compression stress. After absorbing water/blood, the deformed microchannels recovered to their original shapes, and the size of the microchannels had no obvious change before and after absorbing water and blood (Supplementary Fig. 4a–c). Furthermore, a large number of RBCs are aggregated on the surface of the microchannels. The deformed micropores of the ACS recovered to their original state after absorbing water. However, they did not recover to their original shape after contact with blood, and almost no RBCs were observed within the ACS (Fig. 4h).

**In vitro pro-coagulant ability of the MACSs.** We also assessed the pro-coagulant ability of the gauze, GS, CELOX™, CELOX™-G, ACS, MCS-2, and MACSs by the BCI test, in which the lower the BCI value, the stronger the pro-coagulant ability. The BCI values of the MACSs decreased as the porosity increased at 5 and 10 min (Fig. 5a), indicating a positive correlation between the promotion coagulation ability and porosity. The BCI values of the MACSs were significantly lower than that of the ACS (Fig. 5a). Also, the MACS-2 exhibited stronger pro-coagulant ability than the MCS-2 due to the introduction of alkyl chains[24–26]. Notably, the MACSs demonstrated better pro-coagulant performance compared with gauze, GS, CELOX™, and CELOX™-G because of the synergistic effects of the microchannel structure, CS itself, and hydrophobic alkyl chains.

The active coagulation cascade mainly relied on the aggregation of RBCs and adhesion and activation of platelets[5]. Thus, we further evaluated the blood coagulation effect of various samples using RBCs and platelets adhesion assays. The number of adhered RBCs and platelets to the MACSs was higher than that on the gauze, GS, CELOX™, CELOX™-G, ACS, and MCS-2 (Fig. 5b, c). In addition, the higher porosity resulted in a higher number of adhered RBCs and platelets. Consistently, as observed in SEM images, more RBCs and platelets adhered to the MACSs than on other samples (Fig. 5d, e). A higher number of aggregated RBCs was detected in the MACSs than that in other samples (Fig. 5d). Moreover, more activated platelets were observed in the MACS-2 group compared to other groups (Fig. 5f), which accelerated blood coagulation[30]. CS has been proven to accelerate platelet adhesion and activation, and the aggregation of RBCs through electrostatic interactions[31,32]. The microchannel structure was able to promote penetration of the blood and aggregation of RBCs and platelets. The hydrophobic alkyl chains could insert into membranes of RBCs and platelets, further promoting active capture and aggregation[24,25,33]. We concluded that the CS, microchannel structure, and hydrophobic alkyl chains

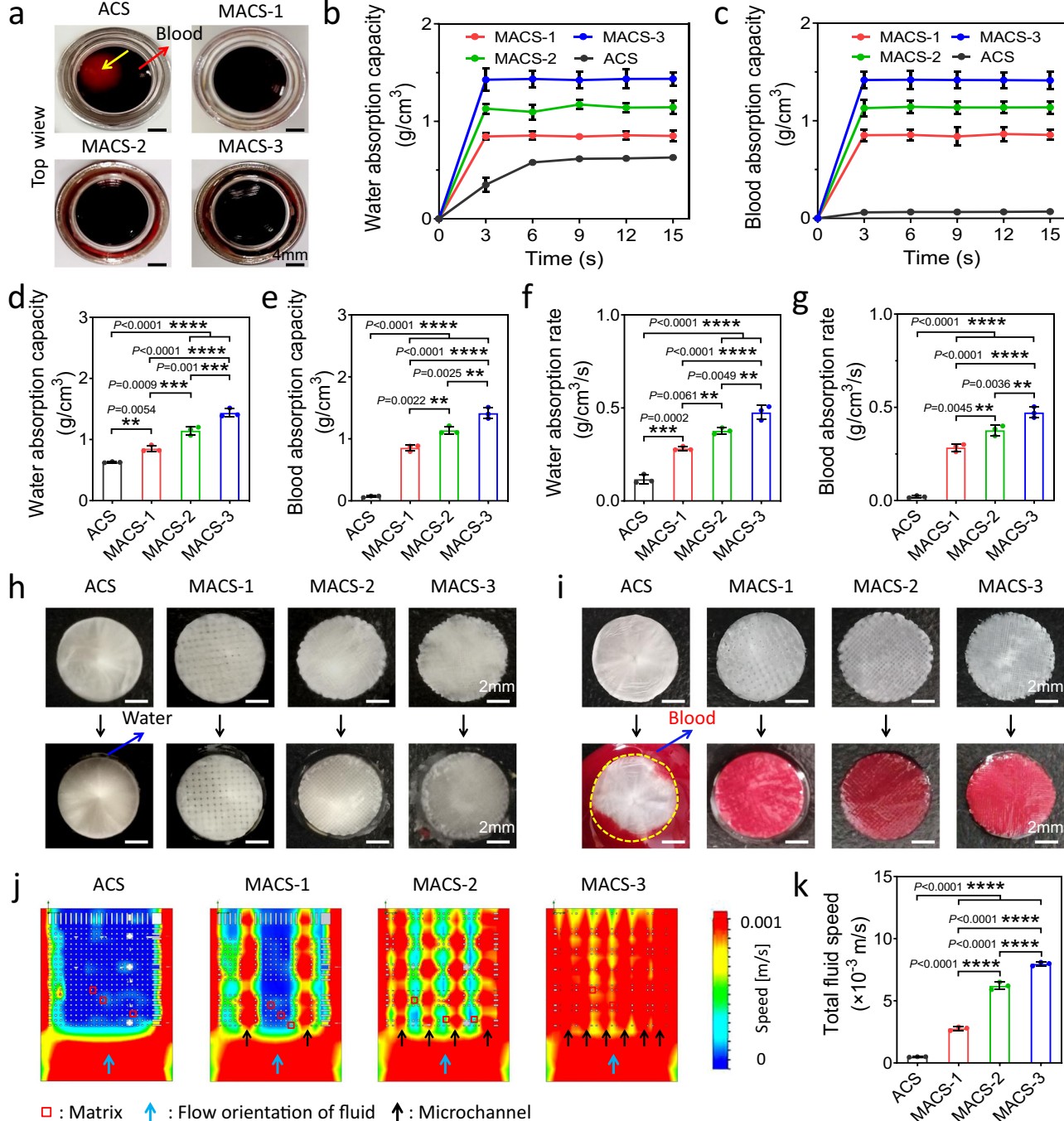

**Fig. 3 Water/blood absorbability of the ACS and MACSs. a** Macro photograph of the ACS and MACS-1/2/3 after absorbing the blood. Yellow and red arrows represented the ACS and blood, respectively. **b, c** Water/blood absorption capacity-time dynamic curves of the ACS and MACS-1/2/3. **d–g** Water/blood absorption capacity and rate of the ACS and MACS-1/2/3. **h, i** Macro photographs of the compressed ACS and MACS-1/2/3 before and after contact with water and blood. The yellow dotted circle represented the boundary of the ACS. **j** Fluid simulation images of water absorption behaviors of the ACS and MACS-1/2/3. **k** Total fluid speed of the ACS and MACS-1/2/3. $n = 3$ independent samples. Data are expressed as mean ± SD. The significant difference was detected by one-way ANOVA with Tukey's multiple comparisons test. The 'ns' indicated no significant difference, **$P < 0.01$, ***$P < 0.001$, ****$P < 0.0001$.

synergistically contributed to the strong pro-coagulant ability of the MACSs (Fig. 5g).

**In vivo hemostatic effect of the MACS-2.** The MACS-2 was selected and used for in vivo hemostasis based on its mechanical strength, water/blood absorbability, blood-triggered shape-memory property, and pro-coagulant capacity (Supplementary Fig. 5).

The hemostatic effect was explored in the normal rat liver perforation wound model, as illustrated in Fig. 6a. After treating the wound with the MACS-2, a small area of bloodstain was observed on the surface of the filter paper beneath the liver, while a large area of bloodstain was sighted in the gauze, GS, CELOX™-G, CELOX™, ACS, and MCS-2 groups (Fig. 6b and Supplementary Movie 9). Quantitatively, the total blood loss of the MACS-2 group was significantly lower than that of other groups (Fig. 6c).

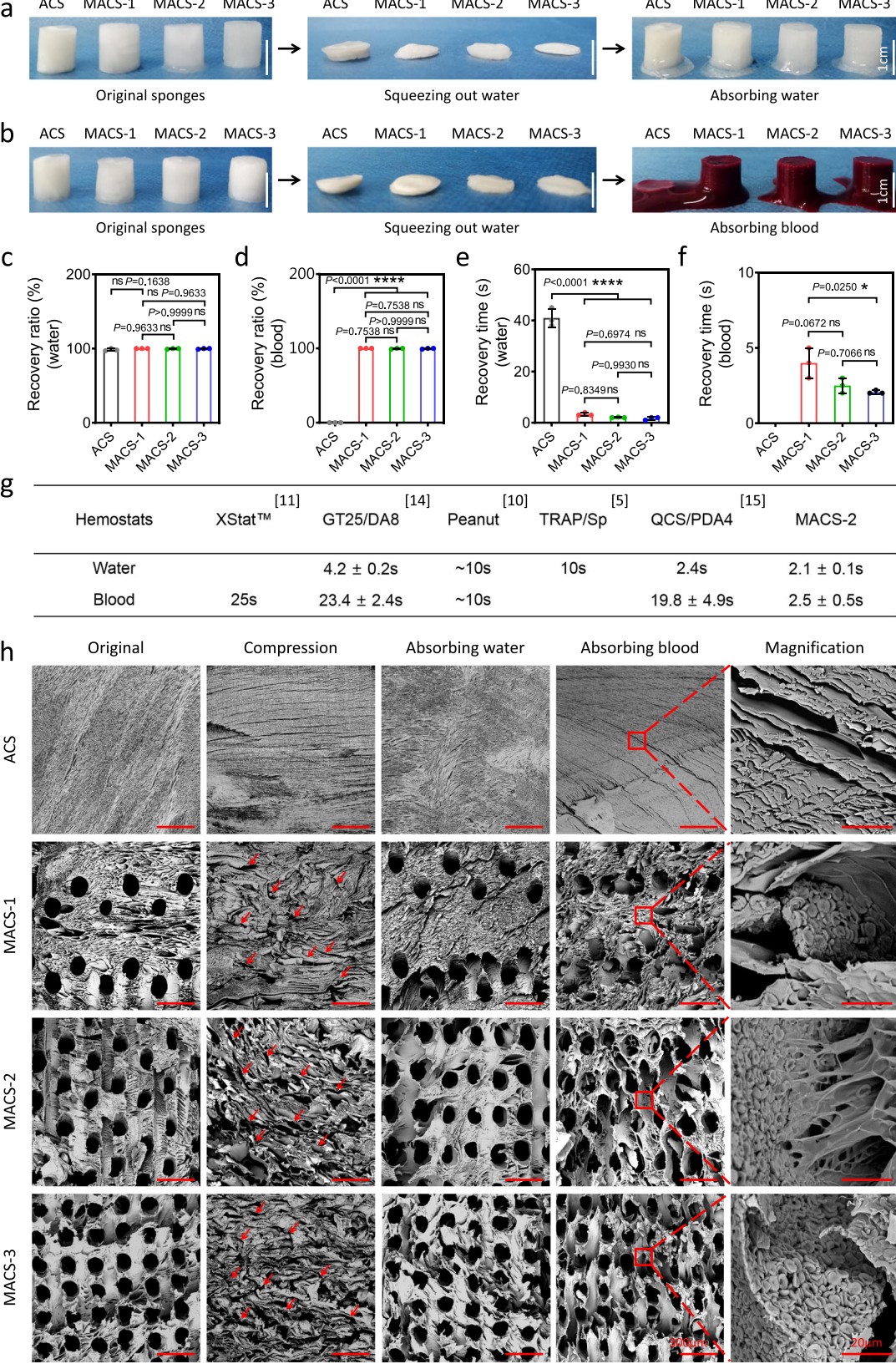

**Fig. 4 Shape-memory property of the ACS and MACSs after absorbing water and blood. a, b** Macro photographs of the water-triggered and blood-triggered shape recovery of the ACS and MACS-1/2/3. **c–f** Shape-recovery ratio and time of the compressed sponges. The shape-recovery time of the ACS was not shown as the compressed ACS could not restore to its original shape after absorbing the blood. $n = 3$ independent samples. Data are expressed as mean ± SD. the significant difference was detected by one-way ANOVA with Tukey's multiple comparisons test. The 'ns' indicated no significant difference, *$P < 0.05$, ****$P < 0.0001$. **g** Comparison of shape-recovery time between the MACS-2 and reported hemostats. **h** SEM images showing the microstructure of the compressed sponges before and after absorbing water and blood. The red arrow represented the deformed microchannel.

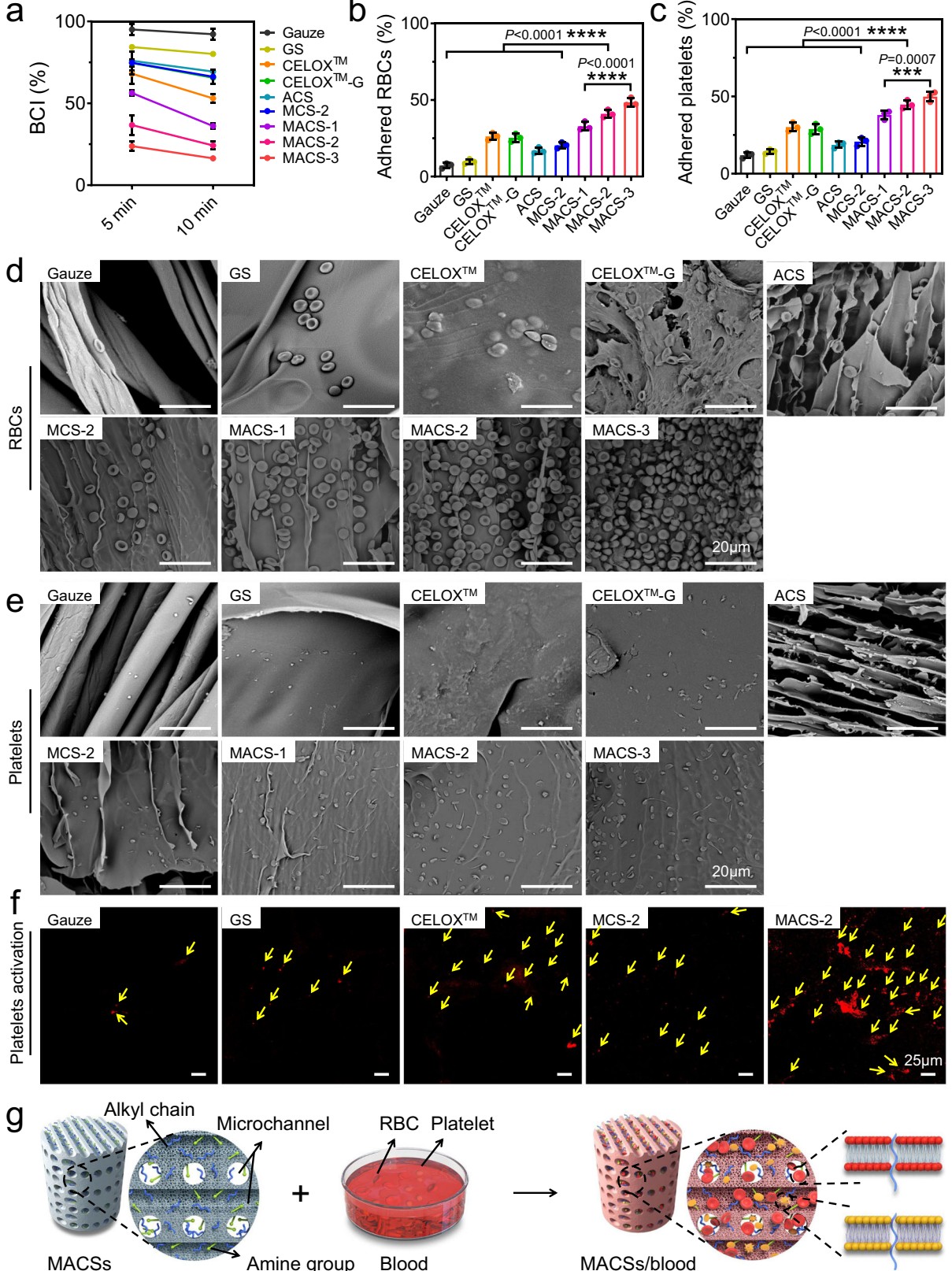

**Fig. 5 The pro-coagulant ability of the gauze, GS, CELOX^TM, CELOX^TM-G, ACS, MCS-2, and MACSs. a** The BCI-time curves of various samples. **b**, **c** The percentage of adhered RBCs and platelets on various samples. $n = 3$ independent samples. Data are expressed as mean ± SD. The significant difference was detected by one-way ANOVA with Tukey's multiple comparisons test. The 'ns' indicated no significant difference, *$P < 0.05$, **$P < 0.01$, ***$P < 0.001$, ****$P < 0.0001$. **d**, **e** SEM images showing adhesion of RBCs and platelets on various samples. **f** Immunofluorescence staining of CD62p showing the activation of platelets on various samples. The yellow arrow represented activated platelet. **g** Schematic diagram illustrating the pro-coagulant mechanism of the MACSs. BCI: blood clotting index; RBCs: red blood cells.

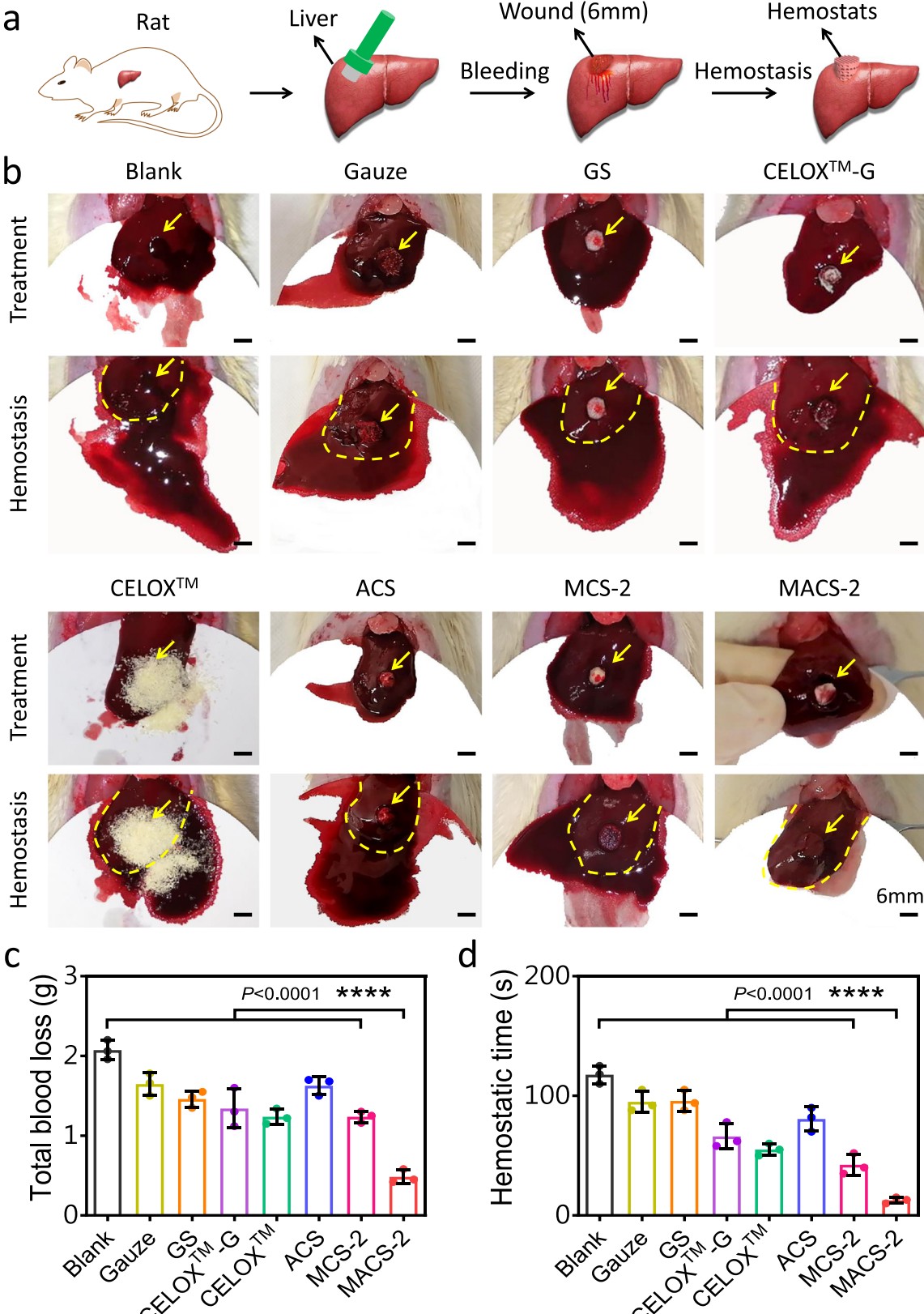

**Fig. 6 Hemostasis in the normal rat liver perforation wound model. a** Schematic illustration of the hemostatic process of hemostats in a rat liver perforation wound model. **b** Photographs of the hemostatic effect of the gauze, GS, CELOX[TM]-G, CELOX[TM], ACS, MCS-2, and MACS-2. The yellow arrow and dotted line represented the bleeding site and liver boundary, respectively. **c**, **d** Total blood loss and hemostatic time in the gauze, GS, CELOX[TM]-G, CELOX[TM], ACS, MCS-2, and MACS-2 groups. $n = 3$ rats per group. Data are expressed as mean ± SD. The significant difference was detected by one-way ANOVA with Tukey's multiple comparisons test. ****$P < 0.0001$.

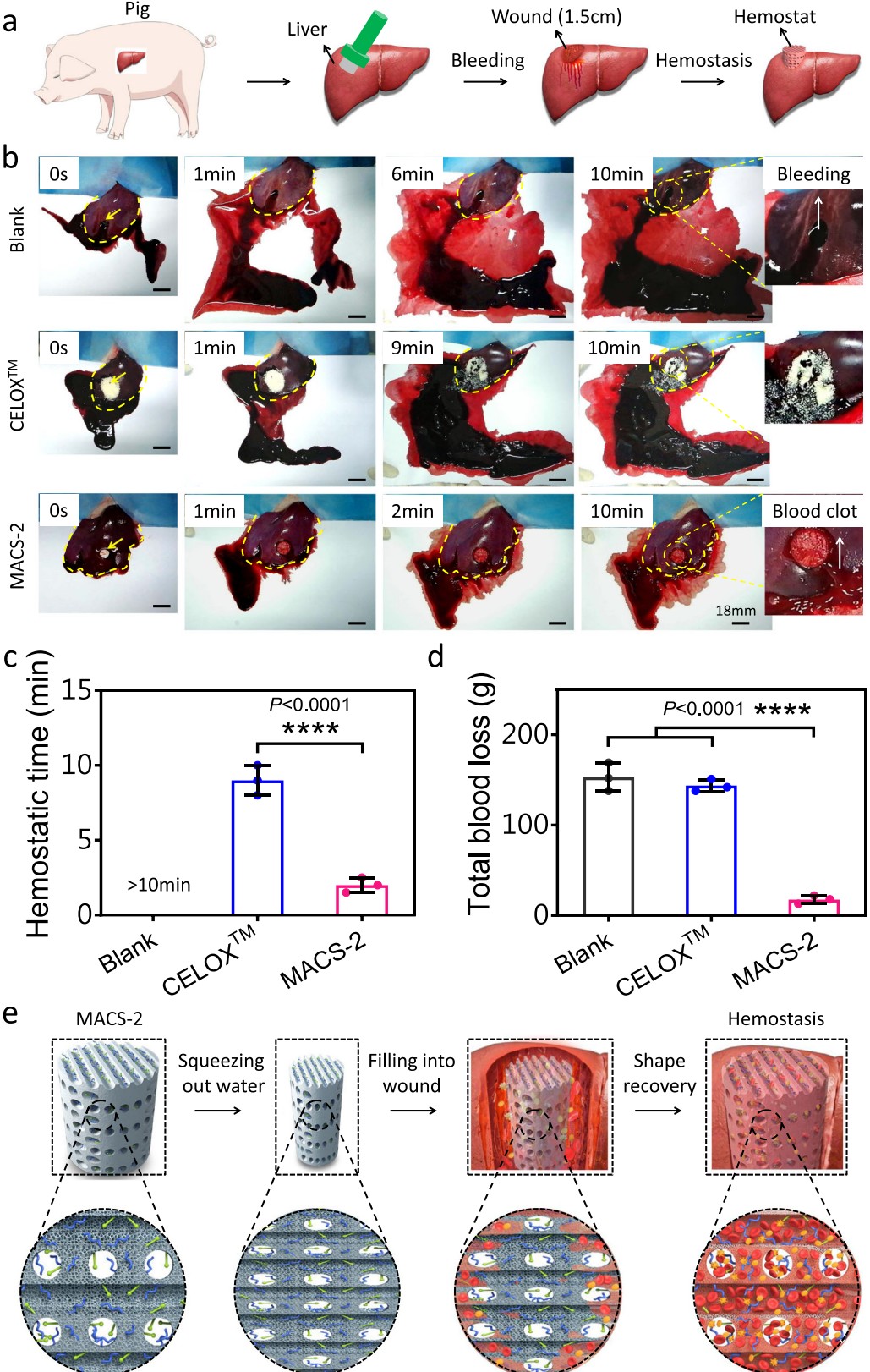

**Fig. 7 Hemostasis in a lethal pig liver perforation wound model. a** Schematic illustration of the hemostatic process of hemostats in a lethal pig liver perforation wound model. **b** Photographs of the hemostatic effect of the blank, CELOX™, and MACS-2 groups. The yellow arrow and dotted line represented the boundary of the liver and the bleeding site, respectively. **c**, **d** Hemostatic time and total blood loss in the blank, CELOX™, and MACS-2 groups. $n = 3$ pigs per group. Data are expressed as mean ± SD. The significant difference was detected by one-way ANOVA with Tukey's multiple comparisons test. ****$P < 0.0001$. **e** Schematic diagram of hemostatic procedure and mechanism of the MACS-2.

Also, the hemostatic time was significantly shorter than that of other groups (Fig. 6d).

Hemorrhage control of anti-coagulated patients remains a challenge in the clinical setting[34]. To simulate clinical application, a heparinized-rat liver perforation wound model was used to evaluate the hemostatic capacity of various samples (Supplementary Fig. 6a). After applying the MACS-2, only a small area of bloodstain was distributed on the surface of the filter paper under the liver (Supplementary Fig. 6b and Supplementary Movie 10). In contrast, a large area of bloodstain was observed after applying other hemostats. Statistical analysis showed that the hemostatic time of the MACS-2 group was much shorter than that of other groups (Supplementary Fig. 6c). Also, the MACS-2 was superior in reducing the total blood loss when compared with the other hemostats (Supplementary Fig. 6d).

To further explore the clinical translational potential of the MACS-2, a lethal pig liver perforation wound model was used to evaluate its hemostatic capacity (Fig. 7a). CELOX™ was used as a control because CELOX™ was a commonly used hemostat in prehospital and hospital scenarios[1,35]. Moreover, similar to the MACS-2, CELOX™ was also made of CS. As the shape-fixed MACS-2 was filled into the wound cavity, it rapidly recovered to its initial shape by absorbing the blood, and then filled the cavity and exerted pressure on the wound wall, achieving hemostasis within $2.0 \pm 0.5$ min (Fig. 7b, c and Supplementary Movie 11). However, the untreated wound continued to bleed for at least 10 min (Supplementary Movie 12), and the CELOX™-treated wound stopped bleeding at $9.0 \pm 1.0$ min (Supplementary Movie 13). The MACS-2 was fixed on the bleeding cavity by its shape recovery. In contrast, the CELOX™ was prone to be washed away by the blood without external compression. In fact, manual pressing is very inconvenient in emergencies and it is difficult for the wounded to complete self-rescue on the battlefield[30]. We further quantified the total blood loss by determining the sum of the weight of the blood absorbed by the filter paper and hemostat. The total blood loss ($17.6 \pm 4.5$ g) in the MACS-2 group was much lower than that in untreated ($153.0 \pm 15.2$ g) and CELOX™ ($143.0 \pm 6.6$ g) groups (Fig. 7d). The MACS-2 demonstrated superior in vivo hemostatic ability for lethal noncompressible hemorrhage compared to clinically used gauze, GS, CELOX™, and CELOX™-G, which was due to the synergistic effect of CS itself, microchannel/microporous structure, and hydrophobic alkyl chains (Fig. 7e). The highly interconnected microchannel structure increased the blood adsorption capacity of the sponge, allowed the blood to perfuse into the interior of the sponge quickly, and then facilitated the recovery of its original shape, which pressed the wound and achieved rapid hemostasis. Moreover, blood cells in high-viscosity blood were difficult to penetrate the interior of matrix micropores, thus, small-size microporous could absorb water in the blood and concentrate blood cells, plasma protein, and coagulation factors in the microchannel, thereby accelerating blood clotting. CS and alkyl chains actively captured RBCs and platelets via intensive interactions, and also promoted aggregation of RBCs and platelets activation. This action triggered the coagulation cascade reaction by fibrinogen-mediated interaction with the activated platelet integrin glycoprotein IIb/IIIa, further improving hemostasis efficiency[1,9,23]. In addition, when the MACS-2 (as a foreign body material) contacted injured vascular tissue, factor XII (Hageman factor in the plasma) was activated and transformed to factor XIIa, triggering an intrinsic coagulation pathway and accelerating fibrin network formation and blood clotting[36].

In war trauma, the rupture of large blood vessels is a common phenomenon, leading to high mortality. Based on it, we further explored the hemostatic capacity of the MACS-2 using a pig femoral artery bleeding model. After injecting the shape-fixed

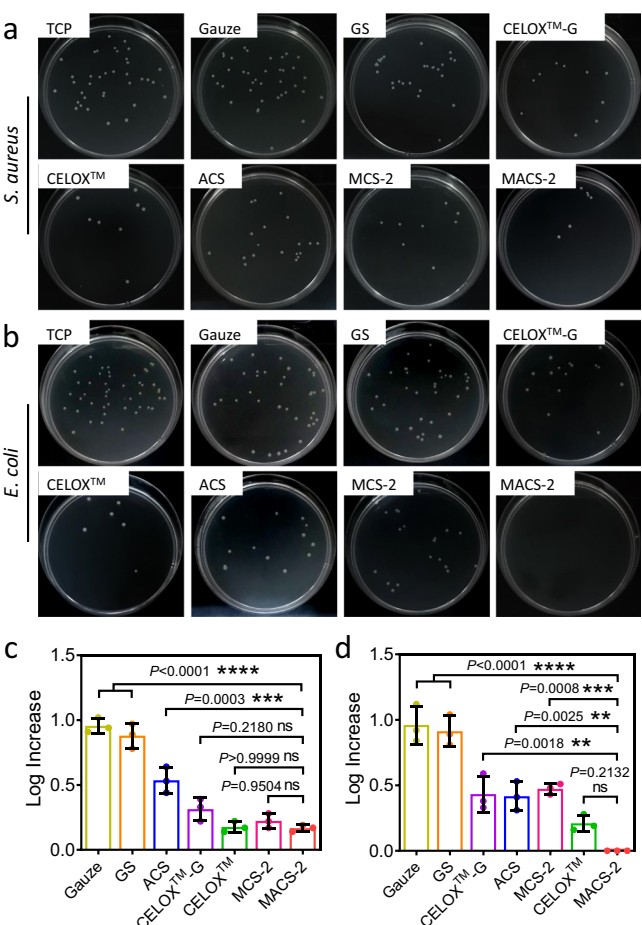

**Fig. 8 In vitro anti-infective property of the MACS-2 and other hemostats. a, b** Photographs of CFUs of *S. aureus* and *E. coli* grown on LB agar plates after contact with TCP, gauze, GS, CELOX™-G, CELOX™, ACS, MCS-2, and MACS-2, respectively. **c, d** Corresponding statistical results of the CFUs of *S. aureus* and *E. coli*. n = 3 independent samples. Data are expressed as mean ± SD. The significant difference was detected by one-way ANOVA with Tukey's multiple comparisons test. The 'ns' indicated no significant difference, **$P < 0.01$, ***$P < 0.001$, ****$P < 0.0001$. *S. aureus*: staphylococcus aureus; *E. coli*: Escherichia coli.

MACS-2 into the wound cavity, the MACS-2 absorbed the blood and recovered to its original shape, thereby pressing the wound and stopping hemorrhage (Supplementary Fig. 7 and Supplementary Movie 14). In addition, the MACS-2 could be customized into different shapes or cut into small pieces to adapt complex wounds with irregular shapes (Supplementary Fig. 8).

**Comparison of in vitro anti-infective property of the MACS-2 with other hemostats.** Severe bacterial infection, similar to massive blood loss, is also responsible for trauma-associated deaths[37]. Thus, ideal hemostats should possess robust anti-infection properties. The anti-infective capacity of the MACS-2 against *S. aureus* and *E. coli* was evaluated by a contact-killing assay and compared with the gauze, GS, CELOX™-G, CELOX™, ACS, and MCS-2 (Fig. 8). Qualitative and quantitative analysis showed that, after contact with the MACS-2, the CFUs number of *S. aureus* was significantly lower than that of the gauze, GS, and ACS groups. There was no obvious difference in the CFUs number between the MACS-2 and CELOX™-G, CELOX™, as well as MCS-2 (Fig. 8a, c). After contact with the MACS-2, the CFUs number of *E. coli* was lower than that of the gauze, GS, CELOX™-G, ACS, and MCS-2 (Fig. 8b, d). There

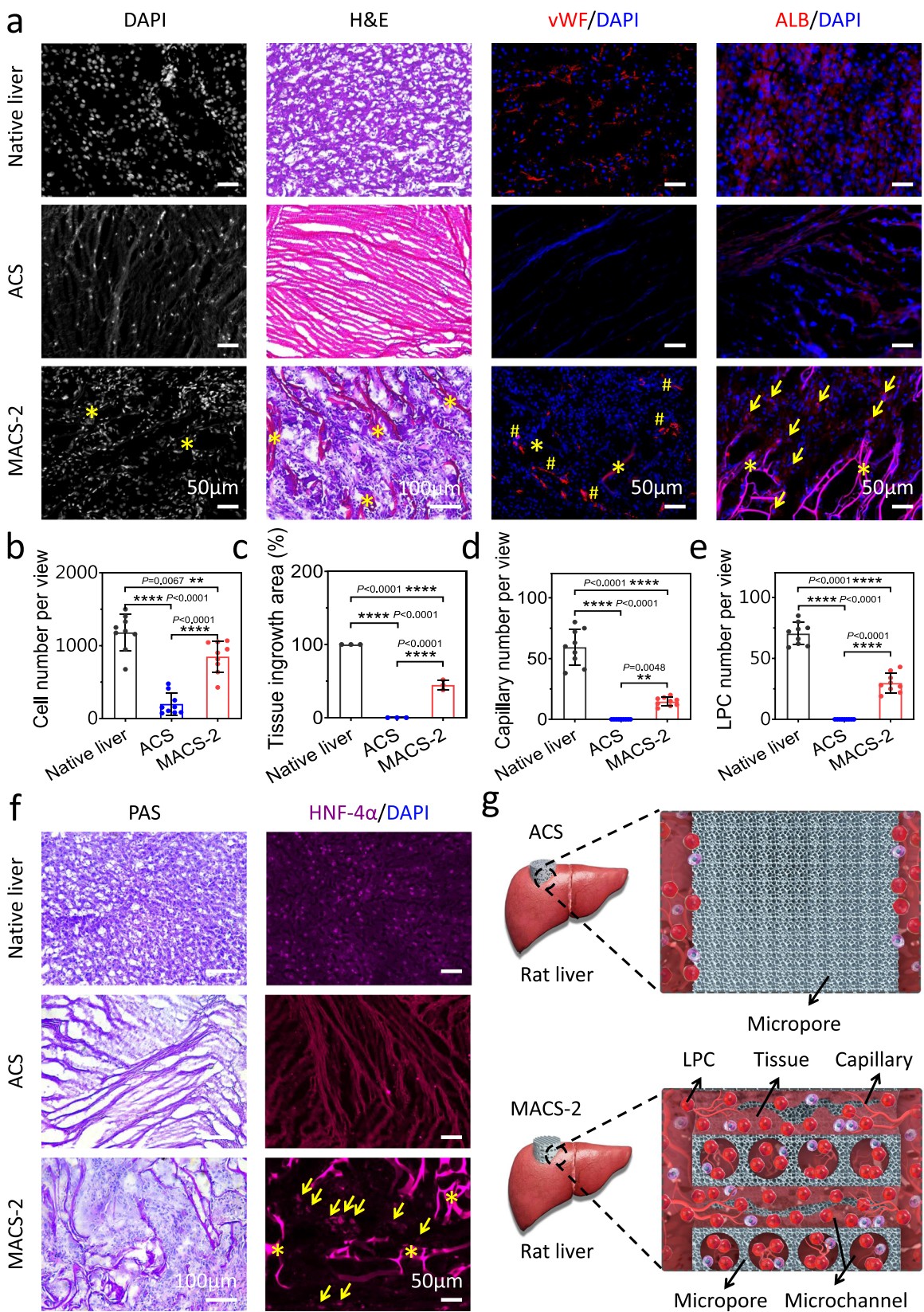

was no significance between the MACS-2 and CELOX™ groups (Fig. 8d). The anti-infective activity of the MACS-2 was ascribed to the synergistic effects of the microchannel structure, grafted hydrophobic alkyl chains, and CS itself. Microchannel structure enabled full contact between bacteria and the MACS-2. The grafted hydrophobic alkyl chains could insert into the lipid bilayer of the bacterial outer membrane and cause bacterial membrane damage, leading to the leakage of intracellular materials[24,26,38]. During

**Fig. 9 Liver regeneration in rat models after implantation of the ACS and MACS-2. a** DAPI staining showing cell infiltration within the ACS and MACS-2. H&E staining showing tissue ingrowth. Images of immunofluorescent staining for vWF (red) and ALB (red) indicating capillary, and LPC infiltration within the ACS and MACS-2. Yellow asterisk, pound key, and arrow represented the alkylated CS, capillary, and LPC, respectively. **b–e** Quantification of cell number, tissue ingrowth area, capillary number, and LPCs within the ACS and MACS-2. $n = 3$ independent samples. Data are expressed as mean ± SD. The significant difference was detected by one-way ANOVA with Tukey's multiple comparisons test. The 'ns' indicated no significant difference, $**P < 0.01$, $****P < 0.0001$. **f** PAS staining showing synthesized hepatic glycogen within the ACS and MACS-2. Images of immunofluorescent staining for HNF-4α (purple) indicating expression of key liver cytokine within the ACS and MACS-2. Yellow asterisk and arrow represented the alkylated CS and HNF-4α, respectively. **g** Schematic illustration of in situ liver regeneration, including the host cell infiltration and vascularization. DAPI: 4′, 6-diamidino-2-phenylindole; LPC: liver parenchymal cell; vWF: von Willebrand factor; ALB: albumin; PAS: periodic acid-schiff; HNF-4α: hepatocyte nuclear factor-4α.

contact with the MACS-2, bacteria generated acidic products (carbonic acid and lactic acid), which yielded a mild acidic microenvironment and further promoted the protonation of amine groups[39,40]. They could react with the negatively charged bacterial membrane by electrostatic interaction, leading to the leakage of proteinaceous and other intracellular constituents of bacteria[24]. Although the MACS-2 was treated by using a NaOH/ethanol solution, residual protonated amine groups also contributed to the antibacterial activity of the MACS-2. Moreover, CS also acted as a chelating agent that selectively bound trace metals and inhibited toxins production and microbial growth[41].

**Hemocompatibility and cytocompatibility of the MACS-2.** Hemocompatibility of the MACS-2 was evaluated and compared with water, PBS, gauze, GS, CELOX™, ACS, and MCS-2. The supernatant in the water group showed bright red, whereas the supernatant in the MACS-2 group presented light pink, which was similar to the color of the supernatant in other groups (Supplementary Fig. 9a). Consistently, the hemolysis ratio of the MACS-2 group was significantly lower than that of the water group (Supplementary Fig. 9b). Although the hemolysis ratios of the MACS-2 and other groups were higher than that of the PBS group, they were much lower than the minimal criteria (5%) of biomaterials hemocompatibility evaluation[42]. Cytocompatibility of the MACS-2 was evaluated by CCK-8 and Live/Dead staining assays. The $OD_{450 nm}$ value of the MACS-2 group gradually increased with the increase of culture time (Supplementary Fig. 9c), which was comparable with that of the GS group. Moreover, almost no dead cells (represented by red signal) were observed in both MACS-2 and GS groups within the 5-day culture (Supplementary Fig. 9d). These results confirmed that the MACS-2 possessed good hemocompatibility and cytocompatibility, which was consistent with other previous studies[24,26,36].

**MACS-2 guided in situ liver regeneration.** The removal of hemostats may result in secondary bleeding and cause great distress to patients. If hemostats could be left in the injury site and directly guide in situ tissue regeneration, this would be favorable to patients and surgeons[9]. In situ liver regeneration as a representative model was used to evaluate the pro-regenerative ability of the MACS-2 and ACS. Rapid host cell infiltration was the first and crucial step for endogenous tissue regeneration[43,44]. DAPI and H&E staining showed that the host cells migrated into the interior of the MACS-2, but were mainly distributed around the edge of the ACS due to its dense structure (Fig. 9a). Accordingly, the cell number inside the MACS-2 was significantly higher than that of the ACS (Fig. 9b). Infiltrated cells secreted a large amount of extracellular matrix and formed neotissue. The tissue ingrowth area within the MACS-2 was much larger than that of the ACS. However, almost no neotissue grew inside the ACS (Fig. 9a, c). A rich capillary network capable of delivering adequate oxygen and nutrients is indispensable for newly formed

tissue survival. Thus, vascularization was assessed by immunostaining for von Willebrand Factor (vWF). A high density of capillaries is distributed inside the MACS-2 (Fig. 9d). In contrast, almost no capillary was observed within the ACS. A large number of ALB-positive cells were observed in the interior of the MACS-2, indicating ingrowth of liver parenchymal cells and liver tissue regeneration. In comparison, almost no liver parenchymal cells infiltrated into the ACS (Fig. 9a, e)[45]. Hepatic glycogen, as an important component of the hepatic cells, plays an essential role in maintaining the relative stability of blood glucose levels. Hepatic glycogen was distributed inside the MACS-2, however, almost no hepatic glycogen was observed within the ACS (Fig. 9f). HNF-4α is a key liver cytokine, which plays an important role in the differentiation and maturation of hepatocytes and liver development. A high density of HNF-4α was observed within the MACS-2, nevertheless, almost no HNF-4α could be detected inside of the ACS (Fig. 9f). The improved ability of cellularization, vascularization, tissue ingrowth, hepatic glycogen synthesis, and expression of HNF-4α of the MACS-2 attributed to the highly interconnected microchannels, high porosity, and good biocompatibility (Fig. 9g)[42]. To our knowledge, there has not been any report to date regarding the use of a shape-memory hemostatic sponge for internal penetrating wound repair. The MACS-2 possessing wet shape-memory property could simultaneously achieve hemostasis and in situ tissue regeneration, which broadens the application of hemostats and opens up an opportunity for the design and construction of clinically beneficial hemostats. Specifically, the application of our MACSs will reduce patient discomfort, simplify treatment procedures, and potentially decrease healthcare costs. In spite of that, the degradation speed of the MACSs was slow, which may be hinder tissue regeneration. Selecting or developing a material with a suitable degradation speed will be an effective solution.

## Methods

**Fabrication of the MACSs.** Chitosan (CS, molecular mass of ~100 kDa) was from Jinan Haidebei Biotech Co., Ltd., China. Dodecyl aldehyde (DA) and sodium cyanoborohydride (NaCNBH₃) were from Shanghai Aladin Co., Ltd., China. Polylactic acid (PLA) filament was from Jinluotuo Biotech Co., Ltd., China. Acetic acid, dichloromethane, and ethyl alcohol were from Tianjin Reagent Co., Ltd., China. All chemicals were of analytical grade.

The fabrication of the MACSs was as follows: First, the PLA microfiber templates with filling ratios of 20, 40, and 60% were printed using a 3D printer (Shenzhen Creality 3D Tech Co., Ltd., China). Second, the templates were filled with CS solution (1, 2, and 4%, w/v) dissolved in acetic acid aqueous solution (2%, v/v), followed by freezing in liquid nitrogen and lyophilization. Third, the CS sponges with microchannel structure were obtained by leaching out the templates with dichloromethane. Residual acetic acid was neutralized with a mixed solution of ethyl alcohol/NaOH (9/1, v/v). The resultant CS sponges were further modified with DA in the presence of NaCNBH₃. Unreacted DA and NaCNBH₃ were removed by rinsing with ethyl alcohol and deionized water (DIW) in turn. The MACSs generated from PLA microfiber templates with filling ratios of 20, 40, and 60% were named as the MACS-1, MACS-2, and MACS-3, respectively. An unmodified micro channeled CS sponge generated from a PLA microfiber template with a ratio of 40% was abbreviated as the MCS-2. The alkylated CS sponge prepared by direct freeze-drying was named the ACS.

**FTIR test**. The spectra of CS powder, PLA microfiber template, PLA/CS composite, and micro channeled CS sponge were recorded in the range of 4000–500 cm$^{-1}$ by using a Fourier transform infrared spectrometer (FTIR, TENSOR II, Germany).

**XPS analysis**. The superficial chemical structure and element content of the CS sponges with or without modification was detected using an X-ray photoelectron spectrometer (XPS, Axis Ultra DLD, England). The N1s peak was treated with CasaXPS software (Version: 2.3.14).

**Characterization of macro/microstructure and porosity**. The macro and microstructure of the MACSs and ACS were characterized by Bruker SkyScan Micro-CT (SkyScan 1276, Allentown, PA, USA) and scanning electron microscopy (SEM, Phenom Pro, Netherlands)[19]. The average pore size was measured using Image-J software (Version: 1.44p). The porosity was calculated using Bruker SkyScan Micro-CT.

**Mechanical test**. The mechanical strength of the MACSs generated with different CS concentrations (1, 2, and 4%, w/v), different PLA microfiber diameter (200 and 400 μm), and PLA microfiber filling ratios (20, 40, and 60%) were prepared into cylindrical shapes (5 mm in height and 8 mm in diameter) and tested in a universal mechanical tester (Instron 3345). The compression strain and speed were fixed at 70% and 1 mm/min, respectively. The maximum compressive stress was obtained from the stress-strain curve. The compressive stress of the MACSs (5 mm in height and 8 mm in diameter) after absorbing the blood was also measured.

**Water/blood absorption behavior**. After squeezing out water, the compressed MACSs and ACS contacted the blood. Their positions in the blood were recorded by a digital camera. To quantitatively evaluate absorption behavior, the volume of MACSs and ACS was measured, called $V$ (cm$^3$), and then the compressed MACSs and ACS were weighed, called $W_d$ (g). After that, the compressed MACSs and ACS were soaked into water and blood from rats. At different time intervals, they were taken out and weighted, called $W_w$ (g). The water/blood absorption capacity was calculated according to the following equation:

$$\text{Water/blood absorption capacity (g/cm}^3) = \frac{(W_w - W_d)}{V} \quad (1)$$

Water/blood absorption rate (g/cm$^3$/s) was calculated by measuring the slope of the water/blood absorption capacity-time curve within 3 s.

Moreover, the absorption behavior was further measured by digital fluid simulation. The MACSs and ACS were modeled by using the Solidworks Flow Simulation software (Solidworks premium 2016 × 64 edition, SolidWorks Corp., MA, USA). The flow orientation of water with a dynamic viscosity of $1.7912 \times 10^{-3}$ Pa. s was parallel to the axial direction of the sponges. The working temperature and pressure were set as 273.2 K and 101325 Pa, respectively. The mass flow at the inlet was 0.001 m/s. To simplify the simulation process, the matrix micropore was replaced by a microchannel. The stimulated microchannel size in the MACSs was about 200 μm.

**Shape-memory property**. The shape-memory property of the MACSs and ACS was evaluated. The MACSs and ACS were compressed to squeeze out free water, and achieved shape fixation. Next, the shape-fixed MACSs and ACS were contacted with water or blood. The shape recovery process was recorded by a digital camera. The shape recovery ratio and time were measured. Also, the microstructure recovery of the MACSs and ACS before and after absorbing water and blood was further observed by SEM. The size of the microchannel was measured with Image-J software.

**Blood clotting index test**. The pro-coagulant ability of the MACSs was evaluated by measuring the blood clotting index (BCI)[29,46]. Gauze, gelatin sponge (GS), CELOX™, CELOX™-gauze (CELOX™-G), ACS, and MCS-2 were used as controls. The MACSs were compressed to squeeze out water and placed in EP tubes. After warming for 10 min at 37 °C, 50 μL of the citrated whole blood (CWB) from rats was dropped onto their top surfaces. After incubation for 5 and 10 min at 37 °C, 3 mL of DIW was added into each EP tube, and optical density value at 540 nm (OD$_{540\,nm}$) of the supernatant was determined using a microplate reader (BIO-RAD, iMARKTM) and called as OD$_{hemostat}$. The mixed DIW/CWB (3 mL/50 μL) solution was used as a negative control and its OD$_{540\,nm}$ value was used as a reference value (OD$_{reference\,value}$). The BCI was calculated based on the following equation:

$$\text{BCI(\%)} = \frac{\text{OD}_{hemostat}}{\text{OD}_{reference\,value}} \times 100\% \quad (2)$$

**RBC and platelet adhesion assays**. The interactions between the MACSs and RBCs were investigated with the previously reported method with some modification[29]. Gauze, GS, CELOX™, CELOX™-G, ACS, and MCS-2 were used as controls. Before the test, RBCs suspension was obtained by centrifuging the CWB

for 10 min under 400×g. The MACSs were compressed to drain off water and placed in a 24-well microplate. Next, 100 μL of RBCs suspension was dropped onto their top surfaces. After incubation for 1 h at 37 °C, they were rinsed with a phosphate buffer solution (PBS, pH = 7.4) to remove nonadherent RBCs, and then transferred into DIW (4 mL) to lyse adhered RBCs to release hemoglobin. After 1 h, 100 μL of the supernatant was taken out and placed into a 96-well microplate followed by measuring its OD$_{540\,nm}$ (OD$_{hemostat}$) value. The OD$_{540\,nm}$ value of a solution composed of 100 μL of RBCs suspension and 4 mL of DIW was used as a reference value (OD$_{reference\,value}$). The percentage of adhered RBCs was calculated by the following equation:

$$\text{RBC adhesion (\%)} = \frac{\text{OD}_{hemostat}}{\text{OD}_{reference\,value}} \times 100\% \quad (3)$$

The interactions between various hemostats and platelets were further evaluated by a platelet adhesion assay[29]. Before measurement, the platelet-rich plasma (PRP) was obtained by centrifuging the CWB for 10 min under 400×g. The MACSs were compressed to squeeze out water and placed into a 24-well microplate. Then, 100 μL of PRP was dropped on their top surfaces followed by incubation for 1 h at 37 °C. Next, they were washed with PBS to remove nonadherent platelets and soaked into a 1% Triton X-100 solution to lyse platelets to release the lactate dehydrogenase (LDH) enzyme. The LDH was determined with an LDH kit (Biyuntian, China) according to its instruction. Finally, the OD$_{490\,nm}$ value of the supernatant was measured and called as OD$_{hemostat}$. The OD$_{490\,nm}$ value of a solution composed of 100 μL of PRP unexposed with hemostats was measured and used as a reference value (OD$_{reference\,value}$). The percentage of adhered platelets was calculated by the following equation:

$$\text{Adhered platelet (\%)} = \frac{\text{OD}_{hemostat}}{\text{OD}_{reference\,value}} \times 100\% \quad (4)$$

The adherence of RBCs and platelets on the various hemostats was observed by SEM. Briefly, hemostats were placed into each well in a 24-well microplate and contacted with 100 μL of RBCs and PRP suspensions. After 1 h at 37 °C, they were rinsed with PBS, and then fixed with 2.5% glutaraldehyde and dehydrated using a series of graded alcohol solutions. After drying, they were cut, and the longitudinal sections were sputtered with gold and observed by SEM. The activated platelets adhering onto the surfaces of gauze, GS, CELOX™-G, CS film, and alkylated CS film were evaluated by immunofluorescence staining for CD62p. The dilutions of Mouse monoclonal anti-P-Selectin (Scbt, sc-8419) and Alexa Fluor 594-conjugated goat anti-mouse IgG (Thermo Fisher Scientific, A11032) were 1:100 and 1:200, respectively. Images were observed and acquired with a laser confocal scanning microscope (Leica, Germany).

**Hemostasis in vivo**. The hemostatic ability of the MACS-2 was evaluated by lethally normal/heparinized rat liver perforation wound models, normal pig liver perforation wound model, and pig femoral artery bleeding model. Gauze, GS, CELOX™, CELOX™-G, ACS, and MCS-2 were used as controls. All animal experiments were performed with the approval of the Animal Experimental Ethics Committee of Nankai University (Protocol number: 2021-SYDWLL-000423).

Normal and heparinized rat liver perforation wound models: A rat (male, weight of 250–300 g, 7–8 weeks) was anesthetized by injecting 10 wt% chloral hydrate in a dose of 1 mL/300 g. Then, the rat's abdomen was incised, and the liver was lifted and placed onto the surface of the preweighted filter paper. Next, a circular perforation wound (diameter of 6 mm) was created on the liver to induce hemorrhaging. Finally, the cylindrical MACS-2 (diameter of 8 mm) was compressed to squeeze out water and filled into the wound cavity. The hemostatic process was recorded with a digital camera. The blood loss was measured by determining the total weight of the blood absorbed by the filter paper and hemostats. The hemostatic time was recorded with a timer. The heparin solution (50UI) was injected into the rat (male, weight of 250–300 g, 7–8 weeks) through a vein at a dose of 2 mL/kg and used for the construction of the heparinized rat liver perforation wound model. Other procedures were similar to the method mentioned above.

Lethal pig liver perforation wound model: Bama miniature pig (male, weight of 15 kg, 3 months) was anesthetized by injecting a mixed solution of midazolam and xylazine hydrochloride (2/1, v/v) into its muscle at a dose of 0.14 mL/1 kg. Then, the abdomen of the pig was incised, and its liver was taken out and placed onto the surface of the filter paper. Next, a 15 mm-diameter circular perforation wound was made on the liver. After bleeding, the cylindrical MACS-2 (diameter of 18 mm) was compressed to squeeze out the free water and filled into the wound cavity. The hemostatic process was recorded with a digital camera. The total blood loss from each liver was weighed and the hemostatic time was recorded.

Lethal pig femoral artery bleeding model: Bama miniature pig (male, weight of 15 kg, 3 months) was anesthetized and fixed. Then, the pig's femoral artery was injured to induce bleeding. Next, the shape-fixed MACS-2 was injected into the wound cavity to stop bleeding. The hemostatic process was recorded with a digital camera.

**In situ liver regeneration**. In situ pro-regenerative ability of the MACS-2 and ACS was evaluated using a representative rat liver defect model. A rat (male, weight of 250–300 g, 7–8 weeks) was anesthetized with 10 wt% chloral hydrate, and its abdomen was incised. Then, a 6 mm-diameter circular perforation wound was created on the liver. Next, the cylindrical MACS-2 was compressed and filled into the wound. As

a comparison, uncompressed ACS was also filled into the wound. After hemostasis, the abdomen was sutured, and the rat was feed normally. After one month post-surgery, the rat was paralyzed, and the liver was taken out for histological and immunofluorescence staining. H&E staining was used to assess tissue ingrowth. DAPI staining was used to evaluate the host cell infiltration. Immunofluorescence staining for von Willebrand factor (vWF) (Abcam, ab6994, 1:100) and albumin (ALB (F-10)) (Scbt, sc-271605, 1:100) was performed to evaluate vascularization and liver par-enchymal cell infiltration. Periodic acid-schiff (PAS) staining was used to assess the synthesis of hepatic glycogen. Immunofluorescence staining for hepatocyte nuclear factor (HNF-4α) (Abcam, ab41898, 1:100) was performed to evaluate the expression of key liver cytokine. The dilutions of Alexa Fluor 594-conjugated goat anti-mouse IgG (Thermo Fisher Scientific, A11032) and Alexa Fluor 594-conjugated goat anti-rabbit IgG (Abcam, ab150080) were 1:200. Images were observed and acquired with the upright microscope (Leica DM3000, Germany) and a fluorescence microscope (Zeiss Axio Imager Z1, Germany).

**In vitro anti-infective activity**. In vitro anti-infective activity of the MACS-2 against *S. aureus* (ATCC6538) and *E. coli* (ATCC25922) was tested by a contact-killing assay with some modification[6,24]. Tissue culture plate (TCP), gauze, GS, CELOX™, CELOX™-G, ACS, and MCS-2 were used as controls. Before the test, the MACS-2 was compressed to squeeze out water and placed into each well in a 48-well microplate. After sterilization for 1 h under UV irradiation, the bacterial suspension (10 μL, $10^8$CFUs/mL) was dropped onto each well. After 2 h at 37 °C, the survival bacteria were resuspended by adding 200 μL of sterilized PBS into each well. Next, 20 μL of resuspended bacterial suspension was taken out and diluted to obtain a final diluting bacterial suspension (FDBS). Subsequently, 20 μL of FDBS was spread onto the surface of the LB agar plate and incubated at 37 °C. After incubation overnight, the formed CFUs on each LB agar plate were counted. The decrease of CFUs was expressed by the following equation:

$$\text{Log increase} = \text{Log survival CFUs of hemostat group} - \text{log survival CFUs of TCP group}$$

$$(5)$$

**Hemocompatibility and cytocompatibility tests**. Hemocompatibility of the MACS-2 was assessed by observing and quantifying the release of hemoglobin. RBCs were obtained by centrifuging CWB from rats at $100 \times g$ for 15 min. Then, RBCs were rinsed with PBS and diluted to 2% (v/v) suspension. Next, RBCs suspension was added into a centrifuge tube to contact with the MACS-2. After incubation for 1 h at 37 °C, the MACS-2/RBCs mixture was centrifuged. A macro photograph of the mixture was collected with a digital camera. The $OD_{540\,nm}$ value of the supernatant was read by a microplate reader. Water, PBS, gauze, GS, CELOX™, ACS, and MCS-2 were used as controls. The hemolysis ratio was calculated based on the following formula:

$$\text{Hemolysis ratio}(\%) = \frac{(OD_h - OD_p)}{(OD_w - OD_p)} \times 100\%$$

$$(6)$$

where $OD_h$, $OD_p$, and $OD_w$ represented the absorbance value of the supernatant in hemostats (gauze, GS, CELOX™, ACS, MCS-2, and MACS-2), PBS, and water groups.

Cytocompatibility of the MACS-2 was evaluated by CCK-8 and Live/Dead staining assays. GS was used as a control. The MACS-2 was immersed into 75% ethanol and washed with PBS. After squeezing out PBS, the MACS-2 was placed into each well in a 48-well plate. Then, 3T3 fibroblast cell suspension ($2 \times 10^4$/well) was dropped onto their top surface. After incubation for 1, 3, and 5 days at 37 °C, the CCK-8 agent was added to each well and further incubated for 4 h. After that, the $OD_{450\,nm}$ value of cell suspension was measured using a microplate reader. In addition, after incubation for 1, 3, and 5 days at 37 °C, a Live/Dead agent was added followed by incubating for 30 min. Next, the stained 3T3 fibroblast cells were observed using a laser confocal scanning microscope.

**Statistics and reproducibility**. All tests were processed in triplicate and similar results were acquired. Each group has three independent samples. Statistical analyses were performed using GraphPad Prism 8 software. Values are expressed as the means ± standard deviation (SD). Comparison between two groups was performed by unpaired two-tailed *t*-test. For multiple group comparison, one-way ANOVA with Tukey's multiple comparison test was used. *$P < 0.05$ was considered to be statistically significant.

**Reporting summary**. Further information on research design is available in the Nature Research Reporting Summary linked to this article.

## Data availability

The relevant data that support the findings of this study are available within the article and its Supplementary Information files or from the corresponding author upon reasonable request. The source data underlying Figs. 1e–n, 2b–d, 3b–g, k, 4c–f, 5a–c, 6c, d, 7c, d, 8c, d, 9b–e and Supplementary Figs. 2, 3c, 4a–c, 6c, d, 9b, c are provided as a Source Data file. Source data are provided with this paper.

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

## Acknowledgements

The authors greatly thank Phillip Bryant for her help on the language revision. This work was financially supported by the National Key Research and Development Program of China (2016YFC1101304), Key Research and Development Program of Ministry of Science and Technology (2017YFC1103500), and National Natural Science Foundation of China (NSFC) projects (31670990, 81921004, 81972063).

## Author contributions

M.Z., L.W., and X.D. conceived the research; X.D., L.W., and H.Y. designed the experiments; X.D., L.W., H.Y., Z.J., Z.C., and S.L. performed the experiments; W.L. and Y.B. characterized the structure of the sponges; X.D., L.W., M.Z., H.W., and D.K. interpreted the data, analyzed the data and wrote the manuscript. All authors discussed the data and direction of the project at regular intervals throughout the study.

## Competing interests

The authors declare no competing interests.
