## [Peer Review File · Nature Communications]

Reviewers' Comments:

Reviewer #1:

Remarks to the Author:

Comments to authors (Manuscript ID: NCOMMS-20-45811)

The authors reported an anti-infective shape-memory hemostatic sponge with ability of guiding in situ tissue regeneration for noncompressible hemorrhage. They designed hemostatic chitosan sponge with highly interconnective microchannels by combining 3D printed fiber leaching and freeze-drying methods, and modified with hydrophobic alkyl chains. The microchannelled alkylated chitosan sponge (MACS) exhibited a strong capacity for water/blood absorption and rapid shape recovery. Compared to clinically used gauze, gelatin sponge, CELOX, and CELOX-gauze, the MACS demonstrated higher pro-coagulant and hemostatic capacities in lethally normal/heparinized rat and pig liver perforation models.

This paper used many methods to characterize MACS. In my opinion, the manuscript can be accepted for publication after major modification.

1. In deep and irregular shape wounds, the resilience of compression sponges usually cause secondary injury to the wound, how to avoid?
2. Water absorption rate is also an important index should be considered. Please provide the experimental data of water absorption rate.
3. What is the design philosophy of the size and diameters of PLA microfiber?
4. Microchannel of MACS is $138\pm 4.3\ \mu\text{m}$ and microporous is $8.7\pm 1.5\ \mu\text{m}$, What is the most appropriate parameter?
5. Does the distribution density of the orifice have any effect on the hemostatic effect?
6. MACS showed reasonable antibacterial activity, what is the antibacterial mechanism? How about bacterial resistance?
7. More experiments, such as, immunohistochemical, expression of key cytokines, are needed to demonstrate the effect on liver tissue integration.

Reviewer #2:

Remarks to the Author:

This manuscript fabricated microchannelled chitosan sponge (MACS) grafting with hydrophobic alkyl chain by using sacrificial PLA microfiber templates printed by a 3D printer. The sponge constructed in this way has well-defined connected microchannels. The MACS has rapid water/blood triggered shape recovery speed, adjusted compression modulus and better hemostatic capacity than gelatin sponge, CELOX, and CELOX-gauze for noncompressible hemorrhage. In addition, it has certain antibacterial properties, and the microporous structure of the MACS promotes liver parenchymal cell infiltration, vascularization, and tissue integration in a rat liver defect model. However, the hemostasis property of the prepared MACS was not better compared to the existing sponges and the manuscript still existed some critical issues in the current form.

1. What is the shape fixation principle of the hemostatic sponge? What is the compression ratio of the hemostatic sponge with fixed shape state? Is there a significant loss in the mechanical properties of the shape fixed hemostatic sponge after recovering to the original shape?
2. Some biochemical tests are needed to prove that the material can promote the activation of blood cells and platelets.
3. In the pig liver punching model, the hemostatic powder used as a control is not a suitable control group.
4. The pig liver punching model is difficult to simulate the bleeding scene and mortality caused by the rupture of large blood vessels in war trauma. It is recommended that the authors add the pig's large arterial bleeding model, such as femoral artery.
5. Chitosan has good antibacterial properties under acidic conditions, but poor antibacterial properties in a neutral environment due to deprotonation. What component does the antibacterial property of hemostatic sponge mainly come from? In the antibacterial statistical results, the data often presented as logarithmic reduction to show the decrease in the number of bacteria, which can better reflect the antibacterial efficiency of the material.
6. The liver is a tissue with strong regeneration ability. The degradation rate of chitosan is slow.

How long does the hemostatic sponge take to degrade during the liver repair? Does the low degradation rate affect the repair of liver tissue? In addition, there is also a lack of a liver damage control group without material.

Reviewer #3:

Remarks to the Author:

Major Revision

The author used a template method to prepare a chitosan-based hemostatic sponge (MACS) with interconnective microchannels. The MACS exhibited capacity for water/blood absorption and rapid shape recovery, good hemostatic performance, and anti-infective activity. It exerts application potential in the field of trauma hemostasis. I think this manuscript needs a major revision before it is considered for publication. A list of specific comments is included:

1. What is the grafting amount of DA? What is the effect of DA graft density on material properties?

2. Fig. 3. The author calculated the water absorption capacity using the equation: liquid absorption weight / material weight (g/g). It would be more appropriate to use liquid absorption weight / material volume (g/cm³).

For ACS, why its blood absorption capacity was significantly lower than water absorption capacity? Please explain.

What is the ordinate unit in Fig. 3G and 3H?

3. Fig. 4. It did not make sense to use a water-saturated sponge for the shape-memory property tests. The shape-memory property and mechanical properties of the aerogels need to be evaluated.

Why the ACS kept a compressed shape when absorbed blood? Please explain or add some details to make it clear.

4. Fig. 5. The authors claimed that the alkyl chains in MACS could insert into cell membranes. To demonstrate the biocompatibility of MACS, hemolysis assay and cytotoxicity assay are necessary.

5. Fig. 6 and 7. The author emphasized the shape-memory property of MACS, which is in favor of stopping bleeding. But, this property was not well used in in vivo hemostatic performance. We could see from hemostatic performance videos and schematic diagram in Fig. 7E, longitudinal shape-recovery ability was very limited. Quantitative data is needed. Swelling characteristics of MACS should be investigated.

Importantly, it seems the MACS could not be used for complex wounds with irregular shapes. Pressing or wound tissue compression will extrude the microchannels, and further, weaken the hemostatic performance of the MACS. A compressed MACS aerogel using in in vivo study will be sufficient to reveal this claim.

To reviewer #1:

Comment 1: In deep and irregular shape wounds, the resilience of compression sponges usually cause secondary injury to the wound, how to avoid?

Response: Thanks for your question. We agree with your professional view. Indeed, the resilience of compression sponges usually caused secondary injury to the wound. After absorbing the blood, the shape-fixed MACS-2 began to recover, thereby filling and pressing the wound (6mm), however, the shape-fixed MACS-2 could not achieve full (100%) shape recovery (8mm) due to the existence of surrounding tissue. The MACS-2 would remain 25% (2mm) compressive strain, whose compressive modulus was just 0.25kPa, which was lower than that (from several kPa to several MPa) of most human soft tissues^{1, 2, 3}. Thus, the MACS-2 did not cause severe pressure and secondary injury to the wound during the application.

[1] Zhao, X., Guo, B., Wu, H., Liang, Y. & Ma, P. X. Injectable antibacterial conductive nanocomposite cryogels with rapid shape recovery for noncompressible hemorrhage and wound healing. *Nat. Commun.* **9**, 2784 (2018).

[2] Serrano, M. C., Chung, E. J. & Ameer, G. A. Advances and applications of biodegradable elastomers in regenerative medicine. *Adv. Funct. Mater.* **20**, 192-208 (2010).

[3] Meyers, M. A., Chen, P. Y., Lin, Y. M. & Seki, Y. Biological materials: Structure and mechanical properties. *Prog. Mater. Sci.* **53**, 1-206 (2008).

Comment 2: Water absorption rate is also an important index should be considered. Please provide the experimental data of water absorption rate.

Response: According to your advice, the experimental data of water absorption rate of the MACSs has been provided in Fig. 3f in the revised manuscript.

Comment 3: What is the design philosophy of the size and diameters of PLA microfiber?

Response: Thanks for your comment. Hitherto, the effect of pore size on hemostatic property has not been systematically investigated. We found that small pore size would inhibit rapid penetration of the blood, and a large pore size indeed accelerated

blood penetration. The PLA microfiber with diameter of 200 and 400 μm was used to construct the MACSs. Both of them showed strong blood absorption ability. However, compared to the MACSs prepared by 200 μm -diameter PLA microfiber, the MACSs prepared by 400 μm -diameter PLA microfiber presented lower mechanical strength (Fig. 1h, i). Low mechanical strength was liable to weaken shape recovery property and hemostatic capacity¹. In addition, large pore size (200 μm) would promote host cell infiltration, vascularization, and tissue ingrowth^{2, 3}. Overall consideration of hemostasis and tissue regeneration, we used 200 μm -diameter PLA microfiber to construct the template.

[1] Yang, X. et al. Peptide-immobilized starch/PEG sponge with rapid shape recovery and dual-function for both uncontrolled and noncompressible hemorrhage. *Acta Biomater.* **99**, 220-235 (2019).

[2] Loh, Q. L. & Choong, C. Three-dimensional scaffolds for tissue engineering applications: role of porosity and pore size. *Tissue Eng. Part B-Re.* **19**, 485-502 (2013).

[3] Lim, K. S., Baptista, M., Moon, S., Woodfield, T. B. F. & Rnjak-Kovacina, J. Microchannels in development, survival, and vascularisation of tissue analogues for regenerative medicine. *Trends Biotechnol.* **37**, 1189-1201 (2019).

Comment 4: Microchannel of MACSs is $138 \pm 4.3\mu\text{m}$ and microporous is $8.7 \pm 1.5\mu\text{m}$, What is the most appropriate parameter?

Response: Thanks for your insightful comment. In fact, microchannel and micropore possessed different functions. Large-size microchannels ($>100\mu\text{m}$) were able to accelerate rapid penetration of high-viscosity blood, and promote host cell infiltration, vascularization, and tissue ingrowth. The size of micropore in matrix was equal to or smaller than that of blood cells¹, and blood cells in high-viscosity blood were difficult to penetrate into the interior of small-size micropore. Therefore, micropore could absorb water in blood and concentrate blood cells, plasma protein, and coagulation factors et al. in the microchannel, thereby accelerating blood clotting.

[1] Li, J et al. Chitosan/diatom-biosilica aerogel with controlled porous structure for rapid hemostasis. *Adv. Healthcare Mater.* **9**, 2000951 (2020).

Comment 5: Does the distribution density of the orifice have any effect on the hemostatic effect?

Response: Thanks for your comment. Hemostatic capacity of the MACSs derived from the synergistic effects of mechanical compression and promotion coagulation, which were associated with the distribution density of the orifice. The high distribution density of the orifice resulted in rapid blood absorption and strong pro-coagulant activity. However, it reduced mechanical strength of the MACSs, which was not conducive for maintaining intact 3D structure and shape of the MACSs under tissue compression and providing long-term hemostatic function.

Comment 6: MACS showed reasonable antibacterial activity, what is the antibacterial mechanism? How about bacterial resistance?

Response: Thanks for your comment. The reasonable antibacterial activity of the MACS resulted from the synergistic effects of grafted hydrophobic alkyl chains and chitosan itself. The grafted hydrophobic alkyl chains could insert into the lipid bilayer of bacterial outer membrane and cause bacterial membrane damage, thereby resulting in bacterial death^{1, 2}. During contact with the MACS, bacteria generated acidic products (carbonic acid and lactic acid), which yielded a mild acidic microenvironment and further promoted the protonation of amine group^{3, 4}. They could react with negatively charged bacterial membrane by electrostatic interaction, leading to the leakage of proteinaceous and other intracellular constituents within bacteria^{1, 2}. Although the MACS-2 was treated by using NaOH/ethanol solution, a small amount of residual protonated amine groups also contributed to anti-bacterial activity of the MACS-2. Moreover, chitosan also acted as a chelating agent that selectively bound trace metals and inhibited the toxins production and microbial growth⁵.

A large number of studies showed that the physical damage to the bacterial cell envelope could limit resistance development⁶. In our study, the antibacterial mechanism of the MACS was mainly the physical damage to the bacterial cell

envelope. Thus, the MACS did not induce bacterial resistance. These discussions have also been supplemented in the revised manuscript.

[1] Chen, G. et al. Wound healing: Bioinspired multifunctional hybrid hydrogel promotes wound healing. *Adv. Funct. Mater.* **28**, 1870233 (2018).

[2] Du, X. et al. Anti-infective and pro-coagulant chitosan-based hydrogel tissue adhesive for sutureless wound closure. *Biomacromolecules* **21**, 1243-1253 (2020).

[3] Hu, J. J. et al. A smart aminoglycoside hydrogel with tunable gel degradation, on-demand drug release, and high antibacterial activity. *J. Control. Release* **247**, 145-152 (2017).

[4] Wang, X. D., Meier, R. J. & Wolfbeis, O. S. Fluorescent pH-sensitive nanoparticles in an agarose matrix for imaging of bacterial growth and metabolism. *Angew. Chem. Int. Edit.* **52**, 406-409 (2013).

[5] Dutta, P. K., Tripathi, S., Mehrotra, G. K. & Dutta, J. Perspectives for chitosan based antimicrobial films in food applications. *Food Chem.* **114**, 1173-1182 (2009).

[6] Makabenta, J. M. V. et al. Nanomaterial-based therapeutics for antibiotic-resistant bacterial infections. *Nat. Rev. Microbiol.* **19**, 23-36 (2021).

Comment 7: More experiments, such as, immunohistochemical, expression of key cytokines, are needed to demonstrate the effect on liver tissue integration.

Response: Thanks for your valuable comment. In order to demonstrate the effect of the MACS on liver tissue integration, we first evaluated the synthesis of hepatic glycogen by periodic acid-schiff (PAS) staining. Hepatic glycogen, as an important component of hepatic cell, played an essential role in maintaining relative stability of blood glucose level. Hepatocyte nuclear factor (HNF-4 α) was a key liver cytokine, which played an essential role in hepatocytes differentiation and maturation and liver development. We also detected the expression of HNF-4 α by immunofluorescence staining. Corresponding results and discussions were added into the revised manuscript (Fig. 9f).

To reviewer #2:

Comment 1: What is the shape fixation principle of the hemostatic sponge? What is

the compression ratio of the hemostatic sponge with fixed shape state? Is there a significant loss in the mechanical properties of the shape fixed hemostatic sponge after recovering to the original shape?

Response: Thanks for your professional comment. The shape fixation principle was the deformation of microchannel and micropore structure within the MACS after squeezing out free water.

The maximum compression ratio of the hemostatic sponge (MACS-2) was about 80% (Fig. 4a, b), which was calculated based on the following formula:

Maximum compression ratio (%)

$$= \left[\frac{\pi \times H_0 \times (D_0/2)^2 - \pi \times H_1 \times (D_1/2)^2}{\pi \times H_0 \times (D_0/2)^2} \right] \times 100\%$$

Where D_0 and D_1 were diameter of the cylindrical sponge at original and shape-fixed states. H_0 and H_1 were height of the cylindrical sponge at original and shape-fixed states.

The mechanical property of the MACS-2 was evaluated by a cyclic compression test. There was almost no loss in the mechanical property of the shape-fixed hemostatic sponge after recovering to the original shape. Result was as follows:

Comment 2: Some biochemical tests are needed to prove that the material can promote the activation of blood cells and platelets.

Response: Thanks for your suggestion. Immunofluorescence staining for CD62p was used to confirm the effect of the MACS-2 on platelet activation. Results were shown in Fig. 5f.

Comment 3: In the pig liver punching model, the hemostatic powder used as a control is not a suitable control group.

Response: Thanks for your professional advice. Hemostatic powder (CELOX™), as a commercial hemostat, has been widely used for hemostasis in prehospital and hospital emergency situations. Moreover, similar with the MACS-2, CELOX™ was also made of chitosan. In fact, we have evaluated hemostatic performance of control sponges (ACS and MCS-2) in rat liver punching model, whose hemostatic capacity was significantly weaker than that of the MACS-2. In addition, there was no commercial shape-memory chitosan-based sponge used for hemostasis of noncompressible penetrating wound. Thus, CELOX™ was used as a control group in pig liver puncturing model. Indeed, as you said, CELOX™ was different in form and application method with the MACS-2. When filled into the wound, CELOX™ needed to be compressed, whereas the MACS-2 did not need manual compression. These results also indicated that the MACS-2 was more suitable for pre-clinical and clinical applications.

Comment 4: The pig liver punching model is difficult to simulate the bleeding scene and mortality caused by the rupture of large blood vessels in war trauma. It is recommended that the authors add the pig's large arterial bleeding model, such as femoral artery.

Response: According to your advice, we evaluated the hemostatic capacity of the MACS-2 in pig femoral artery bleeding model. After injecting shape-fixed MACS-2 into the wound cavity, the MACS-2 recovered to original shape, thereby pressing the wound and effectively stopping hemorrhage. Results were shown in Supplementary Fig. 7 and Supplementary Movie 14 in the revised manuscript.

Comment 5: Chitosan has good antibacterial properties under acidic conditions, but poor antibacterial properties in a neutral environment due to deprotonation. What component does the antibacterial property of hemostatic sponge mainly come from? In the antibacterial statistical results, the data often presented as logarithmic reduction

to show the decrease in the number of bacteria, which can better reflect the antibacterial efficiency of the material.

Response: Thanks for your professional and valuable advice. The antibacterial activity of the MACSs resulted from the synergistic effects of grafted hydrophobic alkyl chains and chitosan itself. The grafted hydrophobic alkyl chains could insert into the lipid bilayer of bacterial outer membrane and cause bacterial membrane damage, thereby resulting in bacterial death^{1, 2}. During contact with the MACS, bacteria generated acidic products (carbonic acid and lactic acid), which yielded a mild acidic microenvironment and further promoted the protonation of amine group^{3, 4}. They could react with negatively charged bacterial membrane by electrostatic interaction, leading to the leakage of proteinaceous and other intracellular constituents within bacteria^{1, 2}. Although the MACS was treated by using NaOH/ethanol solution, residual protonated amine groups also contributed to antibacterial activity of the MACS. Moreover, chitosan also acted as a chelating agent that selectively bound trace metals and inhibited toxins production and microbial growth⁵. These explanations have also been supplemented in the revised manuscript.

In addition, according to your advice, the antibacterial statistical results have been revised and presented in Fig. 8c and d. The decrease in the number of bacteria was calculated by the formula:

$$\begin{aligned} \text{Log Increase} &= \text{Log Survival CFUs of hemostat group} \\ &\quad - \text{Log Survival CFUs of TCP group} \end{aligned}$$

That's because the number of *E. coli* in the MACS-2 group was zero, the decrease in the number of bacteria could not be calculated using the following formula⁶:

$$\begin{aligned} \text{Log Reductio} &= \text{Log Survival CFUs of TCP group} \\ &\quad - \text{Log Survival CFUs of hemostat group} \end{aligned}$$

[1] Chen, G. et al. Wound healing: Bioinspired multifunctional hybrid hydrogel promotes wound healing. *Adv. Funct. Mater.* **28**, 1870233 (2018).

[2] Du, X. et al. Anti-infective and pro-coagulant chitosan-based hydrogel tissue adhesive for sutureless wound closure. *Biomacromolecules* **21**, 1243-1253 (2020).

[3] Hu, J. J. et al. A smart aminoglycoside hydrogel with tunable gel degradation, on-demand drug

release, and high antibacterial activity. *J. Control. Release* **247**, 145-152 (2017).

[4] Wang, X. D., Meier, R. J. & Wolfbeis, O. S. Fluorescent pH-sensitive nanoparticles in an agarose matrix for imaging of bacterial growth and metabolism. *Angew. Chem. Int. Edit.* **52**, 406-409 (2013).

[5] Dutta, P. K., Tripathi, S., Mehrotra, G. K. & Dutta, J. Perspectives for chitosan based antimicrobial films in food applications. *Food Chem.* **114**, 1173-1182 (2009).

[6] Zhao, X., Guo, B., Wu, H., Liang, Y. & Ma, P. X. Injectable antibacterial conductive nanocomposite cryogels with rapid shape recovery for noncompressible hemorrhage and wound healing. *Nat. Commun.* **9**, 2784 (2018).

Comment 6: The liver is a tissue with strong regeneration ability. The degradation rate of chitosan is slow. How long does the hemostatic sponge take to degrade during the liver repair? Does the low degradation rate affect the repair of liver tissue? In addition, there is also a lack of a liver damage control group without material.

Response: We also agree your professional view. Indeed, the liver was a tissue with strong regenerative ability and the degradation rate of chitosan was slow. It was well known that the degradation rate of the scaffold should match with new tissue growth rate. Slow degradation rate would hinder tissue regeneration. After one month post-surgery, the MACS-2 slightly degraded, which was consistent with previous study^{1, 2}. Moreover, previous study indicated that the fully degradation of chitosan needed more than six months³. Selecting or developing a material with suitable degradation rate would be an effective solution.

Taking your advice, a liver damage group without material was used as a control. Two rats were died in untreated group (n=6) due to severe complications caused by massive blood loss, although four rats survived and achieved liver regeneration. Indeed, uncontrolled hemorrhage from noncompressible penetrating wounds resulted in death, and effective hemostasis was a key step for saving life. Moreover, in order to prove the performance of microchannel structure in promoting tissue regeneration, the ACS was also used as a control group.

We hope that you are satisfy with our explanation.

[1] Zhao, X., Guo, B., Wu, H., Liang, Y. & Ma, P. X. Injectable antibacterial conductive nanocomposite cryogels with rapid shape recovery for noncompressible hemorrhage and wound healing. *Nat. Commun.* **9**, 2784 (2018).

[2] Li, M., Zhang, Z., Liang, Y., He, J. & Guo, B. Multifunctional tissue-adhesive cryogel wound dressing for rapid nonpressing surface hemorrhage and wound repair. *ACS Appl. Mater. Interfaces* **12**, 35856-35872 (2020).

[3] Yang, Y. et al. Fabrication and evaluation of chitin-based nerve guidance conduits used to promote peripheral nerve regeneration. *Adv. Eng. Mater.* **11**, B209-B218 (2009).

To reviewer #3:

Comment 1: What is the grafting amount of DA? What is the effect of DA graft density on material properties?

Response: We measured the grafting amount of DA based on XPS analysis. The grafting amount (degree) of DA was $24.67 \pm 13.58\%$ (Fig. 2d).

The grafting density of DA has little effect on microstructure, porosity, mechanical strength, fluid absorbability, and shape recovery property of the MACSs. Reasons are as follows: 1) The microstructure and porosity of the MACSs were mainly determined by the PLA microfiber template and ice crystal; 2) The mechanical strength of the MACSs was associated with the concentration of CS solution, porosity and porous structure; 3) The fluid absorbability of the MACSs mainly depended on microchannel structure and porosity; 4) The shape-memory property of the MACSs was contributed by its microchannel structure, porosity, and fluid absorption rate. Whereas, the grafting density of DA increased hemostatic capacity and anti-infective activity of the MACSs. DA could promote the adhesion and aggregation of red blood cells and platelets on the surface of microchannel wall within the MACSs. DA could insert into bacterial outer membrane and cause bacterial membrane damage, thereby resulting in bacterial death.

Comment 2: Fig. 3. The author calculated the water absorption capacity using the equation: liquid absorption weight / material weight (g/g). It would be more

appropriate to use liquid absorption weight / material volume (g/cm^3). For ACS, why its blood absorption capacity was significantly lower than water absorption capacity? Please explain. What is the ordinate unit in Fig. 3G and 3H?

Response: According to your advice, the water absorption capacity was recalculated by the equation (liquid absorption weight / material volume (g/cm^3)). Original data was replaced by new one. Please see Fig. 3b-g in the revised manuscript.

The blood absorption capacity of the ACS was significantly lower than water absorption capacity, because the blood possesses higher viscosity compared with water, inhibiting the complete penetration into the interior of the ACS with small pore size (Fig. 3i).

The ordinate unit in original Fig. 3g and 3h was $\text{g}/\text{cm}^3/\text{s}$, and replaced by Fig. 3f and g in the revised manuscript.

Comment 3: Fig. 4. It did not make sense to use a water-saturated sponge for the shape-memory property tests. The shape-memory property and mechanical properties of the aerogels need to be evaluated. Why the ACS kept a compressed shape when absorbed blood? Please explain or add some details to make it clear.

Response: In our study, the MACSs was a wet-type shape-memory sponge. Thus, we adopted a wet MACSs for the shape-memory property test. This is a commonly used method for evaluating the shape-memory property of wet-type hemostatic sponge^{1,2,3}. Before test, the free water within the MACSs was squeezed out, and then the MACSs achieved shape fixation. After absorbing water and blood, the shape-fixed MACSs restored to the original shape (Fig. 4a, b).

Following your advice, we tested the shape-memory and mechanical properties of the aerogel (dry MACS-2). The aerogel could not achieve shape recovery after contact with the blood (a), however, the aerogel showed increased mechanical strength compared to wet MACS-2 (b). Thus, it was suggested that the MACSs should be stored and used in the wet state.

The compressed ACS possessed extremely dense porous structure (Fig. 3i and Fig. 4h), which inhibited the penetration of high-viscosity blood into the interior of the ACS (Fig. 3i). Thus, the compressed ACS could not recover to the original shape even after two days.

[1] Huang, Y. et al. Degradable gelatin-based IPN cryogel hemostat for rapidly stopping deep noncompressible hemorrhage and simultaneously improving wound healing. *Chem. Mater.* **32**, 6595-6610 (2020).

[2] Zhao, X., Guo, B., Wu, H., Liang, Y. & Ma, P. X. Injectable antibacterial conductive nanocomposite cryogels with rapid shape recovery for noncompressible hemorrhage and wound healing. *Nat. Commun.* **9**, 2784 (2018).

[3] Yang, X. et al. Peptide-immobilized starch/PEG sponge with rapid shape recovery and dual-function for both uncontrolled and noncompressible hemorrhage. *Acta Biomater.* **99**, 220-235 (2019).

Comment 4: Fig. 5. The authors claimed that the alkyl chains in MACS could insert into cell membranes. To demonstrate the biocompatibility of MACS, hemolysis assay and cytotoxicity assay are necessary.

Response: According to your valuable comments, hemolysis assay and cytotoxicity assay have been carried out, and corresponding results were presented in Supplementary Fig. 9.

Comment 5: Fig. 6 and 7. The author emphasized the shape-memory property of MACS, which is in favor of stopping bleeding. But, this property was not well used in in vivo hemostatic performance. We could see from hemostatic performance videos and schematic diagram in Fig. 7E, longitudinal shape-recovery ability was very limited. Quantitative data is needed. Swelling characteristics of MACS should be investigated. Importantly, it seems the MACS could not be used for complex wounds with irregular shapes. Pressing or wound tissue compression will extrude the microchannels, and further, weaken the hemostatic performance of the MACS. A compressed MACS aerogel using in in vivo study will be sufficient to reveal this claim.

Response: The MACS was not longitudinally compressed due to the shape limitation of cylindrical liver penetrating wound model. In fact, the MACS possessed well longitudinal shape-recovery ability (Fig. 4a, b and Supplementary movies 5, 7). Quantitative data was shown in Fig. 4c, d.

Taking your advice, we tested swelling ratio of the MACS-2 according to the following formula:

$$R_v = \frac{V_t}{V_o}$$

Where V_o and V_t represented the volume of the MACS-2 at original state and swelling

state, respectively. The MACS did not swell after immersion in water for 8 weeks, as shown in following figure.

The MACS could be cut into small pieces to adapt the complex wound with irregular shapes, as shown in following figure.

The shape-fixed MACS did not need manual compression during hemostasis in rat and pig liver penetrating wound models. When the MACS was filled into the wound, it could rapidly recover to original shape by absorbing the blood. After contact with the MACS, the blood occurred coagulation and converted into blood clots. Blood clots within microchannels reinforced the mechanical strength of the MACS and prevented MACS deformation caused by surrounding tissue.

Following your and **Reviewer#2's** suggestions, we supplemented pig femoral artery bleeding model, in which the MACS needed to be manually pressed during hemostasis. Manual compression could not weaken the hemostatic performance of the MACS, which was capable of stopping hemorrhage effectively (Supplementary Fig. 7 and Supplementary Movie 14). In addition, the MACS aerogel (dry sponge) could not recover to original shape after contact with the blood. Therefore, we stored and used MACS in the wet state.

Reviewers' Comments:

Reviewer #1:

Remarks to the Author:

Comments to the Author

All the previous concerns have been properly addressed. This manuscript is acceptable at the current status.

Reviewer #2:

Remarks to the Author:

Authors have fully addressed the concerns raised by reviewers, so this manuscript is recommended for acceptance.

Reviewer #3:

Remarks to the Author:

I have run all the reviewers comments, the authors have revised the manuscript following the suggestions.

Point-by-point response to reviewers' comments

Reviewer #1 (Remarks to the Author):

Comment: All the previous concerns have been properly addressed. This manuscript is acceptable at the current status.

Response: We thank the reviewer's comments in helping us to improve our manuscript.

Reviewer #2 (Remarks to the Author):

Comment: Authors have fully addressed the concerns raised by reviewers, so this manuscript is recommended for acceptance.

Response: We thank the reviewer's comments in helping us to improve our manuscript.

Reviewer #3 (Remarks to the Author):

Comment: I have run all the reviewers comments, the authors have revised the manuscript following the suggestions.

Response: We thank the reviewer's comments in helping us to improve our manuscript.